# Evaluation of the hyperspectral radiometer (HSR1) at the ARM SGP site

Kelly A. Balmes[1, 2], Laura D. Riihimaki[1, 2], John Wood[3], Connor Flynn[4], Adam Theisen[5], Michael Ritsche[5], Lynn Ma[6], Gary B. Hodges[1, 2], Christian Herrera[1, 2]

[1]Cooperative Institute for Research in Environmental Sciences, University of Colorado Boulder, Boulder, CO, USA
[2]Global Monitoring Laboratory, National Oceanic and Atmospheric Administration (NOAA), Boulder, CO, USA
[3]Peak Design Ltd, Sunnybank House, Wensley Rd, Winster, Derbys, DE4 2DH, UK
[4]School of Meteorology, University of Oklahoma, Norman, OK, USA
[5]Argonne National Laboratory, Lemont, IL, USA
[6]Brookhaven National Laboratory, Upton, NY, USA

Correspondence to: Kelly A. Balmes (kelly.balmes@noaa.gov)

**Abstract.** The Peak Design Ltd hyperspectral radiometer (HSR1) was tested at the Atmospheric Radiation Measurement User Facility (ARM) Southern Great Plains (SGP) site in Lamont, Oklahoma for two months from May to July 2022. The HSR1 is a prototype instrument that measures total ($F_{total}$) and diffuse ($F_{diffuse}$) spectral irradiance from 360 to 1100 nm with a spectral resolution of 3 nm. The HSR1 spectral irradiance measurements are compared to nearby collocated spectral radiometers including two multifilter rotating shadowband radiometers (MFRSR) and a shortwave array spectroradiometer—hemispheric (SASHe). The $F_{total}$ at 500 nm for the HSR1 compared to the MFRSRs have a mean (relative) difference of 0.01 W m$^{-2}$ nm$^{-1}$ (1-2%). The HSR1 mean $F_{diffuse}$ at 500 nm is smaller than the MFRSRs by 0.03-0.04 (10%) W m$^{-2}$ nm$^{-1}$. The HSR1 clear-sky aerosol optical depth (AOD) is also retrieved by considering Langley regressions and compared to collocated instruments such as the Cimel sunphotometer (CSPHOT), MFRSRs, and SASHe. The mean HSR1 AOD at 500 nm is larger than the CSPHOT by 0.010 (8%) and larger than the MFRSRs by 0.007-0.017 (6-18%). In general, good agreement between the HSR1 and other instruments is found in terms of the $F_{total}$, $F_{diffuse}$, and AODs at 500 nm. The HSR1 quantities are also compared at other wavelengths to the collocated instruments. The comparisons are within ~10% for the $F_{total}$ and $F_{diffuse}$, except for 940 nm where there is relatively larger disagreement. The AOD comparisons are within ~10% at 415 and 440 nm, however, a relatively larger disagreement in the AOD comparison is found for higher wavelengths.

## 1 Introduction

The shortwave (SW) radiation reaching the surface is dependent on the radiation incident at the top of the atmosphere (TOA) and the aerosols, clouds, and other atmospheric constituents that scatter, absorb, and extinguish the incoming radiation as it passes through the atmosphere. The surface downwelling SW radiation varies spatially, temporally,

and spectrally. By measuring the spectral SW radiation reaching the surface, insight into the physical, microphysical, and optical properties of aerosols and clouds are possible (Riihimaki et al., 2021).

Filter-based spectral SW radiation measurements have provided insight into the spectral characteristics of various atmospheric components by measuring at narrowband channels (Michalsky and Long, 2016; Riihimaki et al., 2021). For example, the multifilter rotating shadowband radiometers (MFRSR) (Harrison et al., 1994; Harrison and Michalsky, 1994; Hodges and Michalsky, 2016) and Cimel sunphotometer (CSPHOT) (Holben et al., 1998; Giles et al., 2019) have increased knowledge on aerosols (e.g., McComiskey and Ferrare, 2016), clouds (e.g., Michalsky and Long, 2016; Min et al., 2008; Wang and Min, 2008), water vapor (e.g., Turner et al., 2016; Michalsky et al., 1995), and trace gases (e.g., Alexandrov et al., 2002a&b). In tandem with the increasing need for further understanding of aerosols and clouds to inform weather, climate, and renewable energy forecasting, spectral SW radiation measurements have advanced and hyperspectral radiometers are more readily available. The Rotating Shadowband Spectrometer (RSS) (Harrison et al., 1999), Shortwave Array Spectroradiometer–Hemispheric (SASHe) (Flynn, 2016), and EKO MS-711 (García-Cabrera et al., 2020) are examples of existing hyperspectral radiometers. However, operations due to rotating shadowbands to measure the diffuse irradiance and calibrations of these instruments are challenging, as good solar alignment is needed for accurate measurements and moving parts have greater potential to fail in the field than stationary instrument components.

In an attempt to ease the operational difficulties of hyperspectral radiometry, a newly developed hyperspectral radiometer with no moving parts and no requirement for rotating shade rings or motorized solar tracking devices is now available called the hyperspectral radiometer (HSR1) (Wood et al., 2017; Norgren et al., 2022). The HSR1 measures total ($F_{total}$) and diffuse ($F_{diffuse}$) spectral irradiance from 360 to 1100 nm with a spectral resolution of 3 nm. The HSR1 optical design is a development of the SPN1, which is a commercially available broadband radiometer (see Wood 1999, Badosa et al., 2014 for a detailed description). The HSR1 is operated by an embedded PC, which also includes measurements of internal pressure and humidity in the case, GPS position, and orientation, and the whole system is built into a rugged case (see Fig. 2 in Wood et al., 2017).

The HSR1 was designed with seven spectral sensors: six sensors placed on a hexagonal grid, one sensor at the centre, under a complex static shading mask (see Figs. 1 in Badosa et al., 2014 and Wood et al., 2017). The shading mask design is to ensure that, at any time, for any location: (1) at least one sensor is always exposed to the full solar beam; (2) at least one sensor is always completely shaded and; (3) the solid angle of the shading mask is equal to $\pi$ thus corresponding to half of the hemispherical solid angle. With no moving parts or specific azimuthal alignment, the instrument is ideal for deployment on moving platforms such as ships and remote locations where regular maintenance is difficult.

Assuming isotropic diffuse sky radiance, the third property related to the shading mask implies that all sensors receive equal amounts (50%) of $F_{diffuse}$ from the rest of the sky hemisphere. Therefore, at any instant, the minimum signal ($F_{min}$) measured among the seven sensors is the shaded sensor, which measures half the $F_{diffuse}$, and the maximum signal ($F_{max}$) from among the seven sensors is fully exposed to the solar beam and, therefore, measures the direct irradiance ($F_{direct}$) plus half the $F_{diffuse}$. From this, the following relationships can be formed:

$$F_{diffuse} = 2F_{min},\qquad(1)$$

$$F_{direct} = (F_{max} - F_{min}),\qquad(2)$$

$$F_{total} = F_{direct} + F_{diffuse} = F_{max} + F_{min}.\qquad(3)$$

In the HSR1, $F_{max}$ and $F_{min}$ are selected from the integrated spectral measurements from each sensor, and these relationships are applied to the corresponding spectral measurements to calculate the $F_{total}$ and $F_{diffuse}$. Due to the nature of the measurements, the $F_{total}$ and $F_{diffuse}$ are measured simultaneously. This is in contrast to rotating shadowband systems which must make the $F_{total}$ and $F_{diffuse}$ measurements separately and, therefore, at different times.

The spectrometer within the HSR1 is a significant improvement over those reported in Wood et al. (2017), which used either an array of low-cost commercial spectrometers, or a fibre switch with a higher specification spectrometer to measure the seven spectral inputs. The current HSR1 uses a custom designed multichannel spectrometer, which images and spectrally disperses the light from the input fibres onto a 2D image sensor, so all channels are measured simultaneously. This significantly improves the measurement resolution, speed, and matching between the channels compared with the earlier implementations. An early version of this system was also used by Norgren et al. (2022).

In this study, the prototype HSR1 is evaluated. The HSR1 was at the Atmospheric Radiation Measurement User Facility (ARM) Southern Great Plains (SGP) site in Lamont, Oklahoma for a two-month test period from May to July 2022. The ARM SGP site is an ideal location to evaluate a new instrument with the collocation of several instruments making similar measurements as a reference to compare with. The reference instruments include two MFRSRs, a CSPHOT, and a SASHe utilized to evaluate the HSR1's ability to measure $F_{total}$ and $F_{diffuse}$ as well as retrieve aerosol optical depth (AOD).

Sect. 2 describes the HSR1 data and general performance as well as other instruments and data sources utilized in the evaluation. Sect. 3 details the HSR1 AOD retrieval methodology. Sect. 4 presents the results of the HSR1 comparison. Sect. 5 briefly discusses post-processing modifications and calibration checks and the resultant implication on the HSR1 data and evaluation results. Sect. 6 presents concluding remarks.

## 2 Data

### 2.1 HSR1

The HSR1 prototype was at the ARM SGP site in Lamont, Oklahoma (36.61 °N, 97.49 °W) from 16 May 2022 to 18 July 2022 for the test period. The HSR1 was located on the guest instrument facility (GIF) at the Central Facility (C1) (Fig. 1). The HSR1 exhibited excellent uptime and near-autonomous data collection over the two-month test period with an uptime of 97.5%. The HSR1 time period sets the time period for the rest of the study. Other measurements (Sect. 2.2) are considered temporally collocated to the HSR1 when observations are within 1 min. A map showing the spatial distribution of the other instruments is shown in Fig. 1, with all instruments separated by 170 m or less.

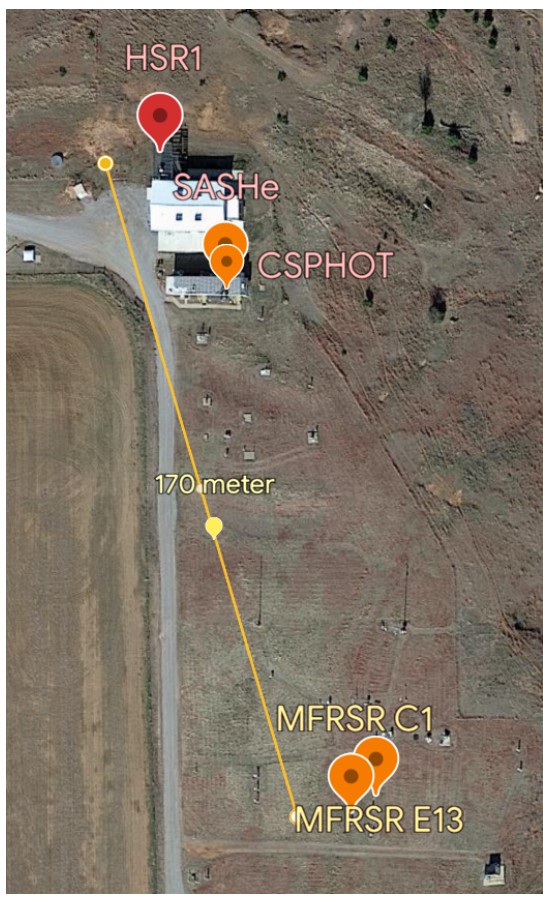

**Figure 1: Map of the instruments at the ARM SGP site. The HSR1 is indicated by a red marker and all other instruments are shown with an orange marker. The instrument names are labelled near their respective markers. The yellow line indicates a distance of 170 m for scale.**

**Table 1. Instrument specifications including spectral range, spectral resolution, retrieved quantities, and uncertainty estimates.**

| Instrument | Measurement | Spectral coverage (resolution) | Retrieved quantities | Uncertainty estimates |
|---|---|---|---|---|
| HSR1 | Total and diffuse hyperspectral irradiances | 360-1100 nm (3 nm) | AOD at 415, 440, 500, 615, 673, 675, and 870 nm | Total irradiances: 5% AOD: 0.02 |
| CSPHOT | Direct solar irradiance and sky radiance | 340, 380, 440, 500, 675, 870, 1020, and 1640 nm | AOD at 440, 500, 675, and 870 nm | AOD: 0.01 |
| MFRSR | Total and diffuse spectral narrowband irradiances | 415, 500, 615, 673, 870, and 940 nm | AOD at 415, 500, 615, 673, and 870 nm | Irradiances: 3% AOD: 0.01 |
| SASHe | Total and diffuse | 336 to 1100 nm (~2.5 nm), 950 | AOD at 415, 500, | AOD: 0.02-0.03 |

| hyperspectral irradiances | to 1700 nm (6 nm) | 615, 673, and 870 nm | Irradiances: AOD relative uncertainty multiplied by the airmass |

The HSR1 spectrometer achieves an optical resolution of 3 nm over the range 350 nm to 1050 nm, and can take a measurement in as little as 200 ms. However, to improve the dynamic range of the instrument over the spectral range, and also capture the range of diurnal irradiances, readings are taken over a series of different integration times, and merged into a single high-dynamic-range measurement. This typically gives a measurement time of around 1 s. There is a trade-off between speed and dynamic range. In this study, measurements were made every 10 s, then averaged and stored every

minute to match common solar radiation datasets.

        Example time series for HSR1 integrated irradiance and example spectra from 11 July 2022 are shown in Fig. 2. The integrated irradiance are the spectral irradiances integrated from 400 to 1000 nm. On this day, the conditions were primarily clear-sky. Other features of note in the time series and spectra from this day will be described throughout the remainder of this section.

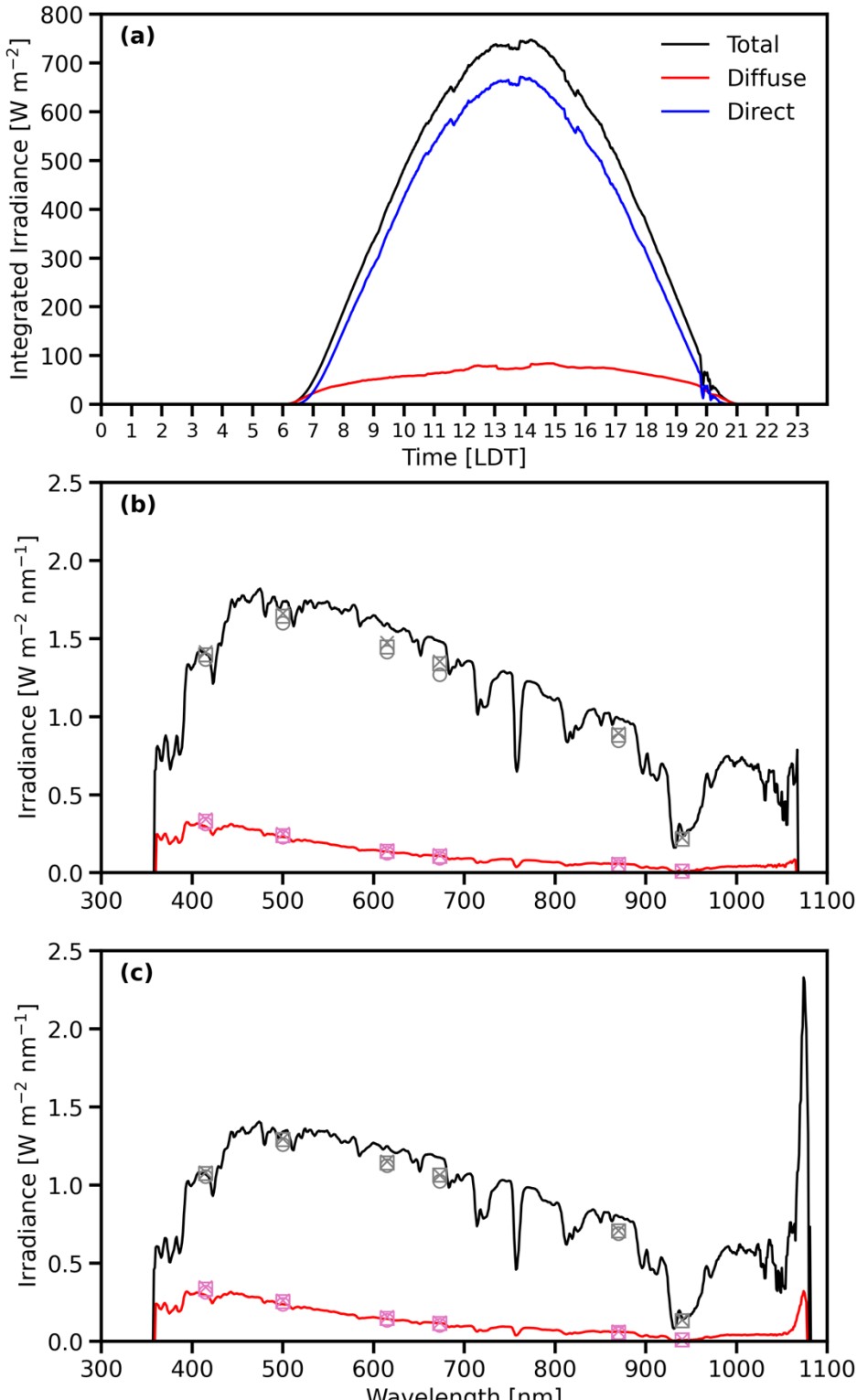

**Figure 2: (a) HSR1 time series, in local daylight time (LDT), of integrated irradiance for $F_{total}$ (black), $F_{diffuse}$ (red), and $F_{direct}$ ($F_{total}$ − $F_{diffuse}$) (blue) irradiances from 11 July 2022. HSR1 spectra are from the same date at (b) 13:53 LDT and (c) 16:28 LDT. Collocated $F_{total}$ (gray) and $F_{diffuse}$ (pink) from the MFRSR C1 (square), MFRSR E13 (x-mark), and SASHe (circle) are also shown.**

115   Several features of the HSR1 performance were noted. The general excellent HSR1 performance is further described in later sections as the main focus of this study. Here, the limited performance issues are described. The data exhibited measurement noise due to straylight issues for wavelengths less than 400 nm and for wavelengths greater than 950 nm. In particular, considerable noise was noted for wavelengths greater than 1000 nm (Fig. 2c) as the measurements were contaminated by second-order straylight as identified in the lab using a monochromator. As with all spectrometers,

120 measurements at the two extremes of the spectrum have low sensitivity and, therefore, additional noise is apparent. Due to the measurement noise, this comparison study focuses on the spectral range of 400 to 950 nm.

   In addition to straylight issues, the data exhibited step functions throughout the diurnal cycle (Fig. 2a). This phenomenon is partially due to the shading mask pattern design as the measurement switches between the seven sensors as the sun angle changes throughout the day. By utilizing seven sensors instead of one sensor, this introduces different

125 calibration errors across the sensors that lead to the step functions. The HSR1 dome also contributes to this issue as the incoming light is bent due to the dome's refracting properties, which is referred to as the dome lensing effect (Badosa et al., 2014). The dome lensing effect can be corrected for by a set of equations that take into account the geometry of the solar position and the HSR1 and the resultant change in angle of the incoming light as the light passes through the dome into the sensors. The dome lensing effect corrected $F_{total}$ and $F_{diffuse}$ are discussed further in Sect. 5.

130   Furthermore, the cosine response of the seven sensor diffusers is measured in the lab using a collimated beam from Xenon lamp, and is within 2% of the normal beam irradiance over the range of zenith angles. The cosine response curves for the HSR1 in this study are shown in Fig. 3.

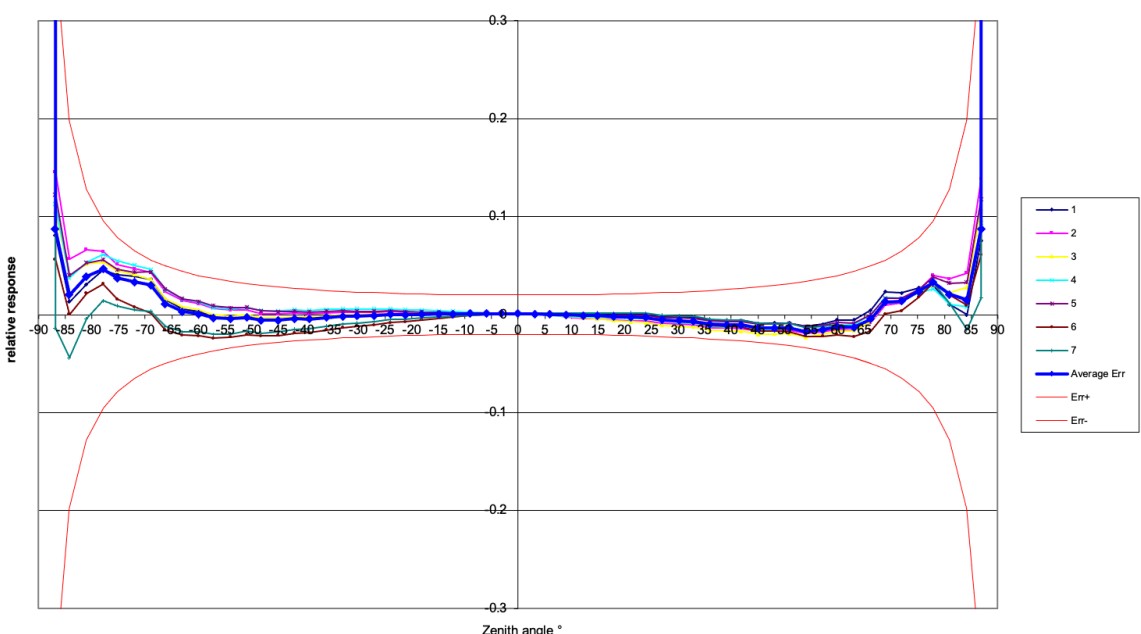

**Figure 3: Cosine curve for the 7 sensors of the HSR1. Heavy blue line shows an average of these, and the red lines show 2% design limits.**

### 2.1.1 Calibration & uncertainty

A reference HSR1 is calibrated by removing the shading mask, and exposing the sensors to a 1000 W 'FEL' lamp, with an output spectrum calibrated by the UK NPL. This calibration is transferred to other HSR1s during routine calibrations and calibration checks using an integrating sphere. The expected uncertainty in $F_{total}$ measurements is expected to be around 5% between 400 nm and 900 nm.

### 2.1.1 Field of View (FOV)

As described in Badosa et al. (2014), the HSR1 optical system has a larger FOV than a typical sun photometer. The precise FOV is somewhat variable, depending on the position of the sun in the sky, but it is typically around ±7°. This means that the circumsolar irradiance will normally be included as part of the $F_{direct}$, rather than in the $F_{diffuse}$, as would be the case with a narrow FOV sun photometer. This means that the HSR1 $F_{diffuse}$ measurement will typically be lower than the corresponding measurements from a sun photometer or broadband tracker system.

## 2.2 Other data

### 2.2.1 CSPHOT

The CSPHOT AODs are considered in the comparison (Holben et al., 1998). The CSPHOT observations are from the Aerosol Robotic Network (AERONET) Version 3 Level 2 AOD data product, which provides quality assured and filtered AODs during clear-sky conditions (Giles et al., 2019). The CSPHOT observations considered include the AODs at 440, 500, 675, and 870 nm. The AERONET AOD uncertainty is 0.01 for the wavelengths considered in this study (Giles et al., 2019)

### 2.2.2 MFRSR

The multifilter rotating shadowband radiometer (MFRSR) measures narrowband $F_{total}$ and $F_{diffuse}$ at 415, 500, 615, 673, 870, and 940 nm (Harrison et al., 1994; Harrison and Michalsky, 1994; Hodges and Michalsky, 2016). Two MFRSRs were collocated to the HSR1 with facility designations C1 and E13. The MFRSR narrowband filters measure with a nominal central wavelength at each desired wavelength and a nominal full width half maximum (FWHM) of 10 nm. The central wavelength and FWHM are measured for each narrowband channel to determine the transmission characteristics of each specific instrument. For example, the MFRSR C1 measured characteristics for the 500 nm channel includes a central wavelength of 501.5 nm and a FWHM of 10.7 nm. The estimated uncertainty in the spectral irradiances is 3%, which is based on the estimated uncertainty of the rotating shadow band spectroradiometer (RSS) that follows the exact same shadowing method (Michalsky and Kiedron, 2023).

In addition to the spectral irradiances, MFRSR-retrieved AODs are also considered at 415, 500, 615, 673, and 870 nm with an estimated uncertainty of 0.01 (Koontz et al., 2013).

### 2.2.3 SASHe

The Shortwave Array Spectroradiometer–Hemispheric (SASHe) measures $F_{total}$ and $F_{diffuse}$ from 336 to 1700 nm (Flynn, 2016) although there are wavelength regions where absorbing gas features hinder radiometric calibration and thus limit the usefulness of the measurement. The spectral resolution of the SASHe is about 2.5 nm for the spectral range where the SASHe and HSR1 overlap. During the course of this study, two instrument issues were identified affecting the operation of the SASHe and the quality of the reported data which limited the data to clear-sky conditions only (see Appendix A). Thus, the SASHe clear-sky $F_{total}$, $F_{diffuse}$, and AODs at 415, 500, 615, 673, and 870 nm are compared to the other instruments.

Due to the data quality issues mentioned above, the SASHe irradiance and AOD uncertainties are difficult to quantify. The uncertainty in AOD is likely not less than 0.02-0.03. The SASHe irradiances are not directly calibrated. Instead, they are derived from Langley calibration where the retrieved TOA spectral irradiance is scaled to agree with those in MODTRAN. Therefore, the uncertainty in the irradiance components and the AOD are directly related. Specifically, the irradiance relative uncertainty will be equal to the relative AOD uncertainty multiplied by the airmass value.

### 2.2.4 RADFLUX

The Radiative Flux Analysis (RADFLUX) data product utilizes quality controlled broadband surface downwelling total ($F_{broadband, total}$) and diffuse ($F_{broadband, diffuse}$) SW irradiance observations to identify clear-sky periods and then calculate clear-sky irradiances (Long and Ackerman, 2000; Long et al., 2006; Riihimaki et al., 2019). The SW broadband radiometer spectral range is 295 to 3000 nm (Andreas et al., 2018). The estimated uncertainties are 4% in $F_{broadband, total}$ and 3% in $F_{broadband, diffuse}$ (Michalsky and Long, 2016).

RADFLUX processing first identifies clear sky time periods using the magnitude and variability of the diffuse and total SW irradiance that have been normalized to remove the impacts of the diurnal cycle. Clear sky estimates are determined at all times using empirical fits to those data points (Long & Ackerman, 2000). Finally, cloud fraction (CF) is calculated based on a relationship with the normalized diffuse cloud effect (i.e., (diffuse measured - diffuse clear sky)/total clear sky). Care is taken to distinguish between optically thin and thick clouds in the CF calculations using statistics on the magnitude and variability of the irradiance measurements and the diffuse ratio (see Long et al. 2006 for more details).

In this study, the clear-sky identified time periods from RADFLUX are considered for the AOD retrieval (Sect. 3) based on when the SW CF is equal to 0. The SW CF uncertainty is 10% (Long et al., 2006). In addition, the broadband diffuse ratio (i.e., $F_{broadband, diffuse}/F_{broadband, total}$) from RADFLUX is compared to that from the HSR1. The RADFLUX data product considered in this study has the facility designation E13.

### 2.2.6 OMI

Ozone column amount for the AOD retrieval is from the ozone monitoring instrument (OMI) on board the Aura satellite (Levelt et al., 2014). Global coverage at a spatial resolution of 1° latitude by 1° longitude of daily ozone values from OMI are provided in the gecomiX1.a1 data product. The daily ozone value corresponding to SGP's latitude and longitude are considered in the HSR1 AOD retrieval (Sect. 3).

### 3 AOD retrieval

The HSR1 AOD is retrieved by considering Langley regressions. The HSR1 AOD retrieval is based on the AOD retrieval methodologies of the MFRSR (Koontz et al., 2013) and SASHe (Ermold et al., 2013). Only clear-sky periods are considered, which are based on the RADFLUX SW CF (Sect. 2.2.4). The AOD are found for wavelengths with corresponding CSPHOT and MFRSR retrieved AODs: 415, 440, 500, 615, 673, 675, and 870 nm.

For a clear-sky atmosphere (i.e., no clouds), the spectral direct normal irradiance (DNI) at a given wavelength ($\lambda$) that reaches the surface can be described as:

$$DNI(\lambda) = DNI_0(\lambda)exp\left[-\left(\tau_{Rayleigh}(\lambda) + \tau_{aerosol}(\lambda) + \tau_{gas}(\lambda)\right)m\right], \tag{4}$$

where $DNI_0$ is the DNI at the top of the atmosphere (TOA), $\tau_{Rayleigh}$ is the Rayleigh optical depth due to molecular

scattering, $\tau_{aerosol}$ is the AOD, $\tau_{gas}$ is the gaseous absorption optical depth, and $m$ is the airmass. By considering the

gaseous absorption as linearly proportional to the airmass and taking the natural logarithm, Eq. 2 becomes:

$$ln\left(DNI(\lambda)\right) = ln\left(DNI_0(\lambda)\right) - \left(\tau_{Rayleigh}(\lambda) + \tau_{aerosol}(\lambda) + \tau_{gas}(\lambda)\right)m \,. \tag{5}$$

By linearly regressing the HSR1 spectral DNI and airmass, the TOA DNI (from the y-intercept) and total optical depth (from

the slope) can be found.

Besides DNI and AOD, the other terms in Eq. 3 are calculated as follows. The Rayleigh optical depth is calculated

as (Hansen and Travis, 1974):

$$\tau_{Rayleigh} = \frac{p}{1013.25} 0.008569\lambda^{-4}\left(1 + 0.0133\lambda^{-2} + 0.00013\lambda^{-4}\right), \tag{4}$$

where $p$ is the surface pressure in mb, and $\lambda$ is the wavelength in microns. The surface pressure considered is from

RADFLUX. The airmass is calculated as (Kasten and Young, 1989):

$$m = \frac{1}{cos\left(\theta_s\right) + 0.50572\left(96.07995 - \theta_s\right)^{-1.6364}}, \tag{5}$$

where $\theta_s$ is the solar zenith angle in degrees.

For $\tau_{gas}$, only the effect of ozone is considered due to the wavelengths considered as other gaseous absorption is

considered negligible (Koontz et al., 2013; Ermold et al., 2013). In addition, only the column amount of ozone is considered

(i.e., no vertical dependence). The ozone optical depth, $\tau_{ozone}$, is calculated as:

$$\tau_{ozone}(\lambda) = \frac{ozone\ columnar\ amount}{1000} x\ A_{ozone}(\lambda), \tag{6}$$

where *ozone columnar amount* is the total amount of ozone in the atmospheric column in Dobson units and $A_{ozone}$ is the

spectrally-dependent ozone gas absorption coefficient. The ozone columnar amounts are from the daily ozone satellite value

(Sect. 2.2.6) that are closest in time and the absorption coefficients are from Ermold et al. (2013) (see their Appendix A).

Note that water vapor is not included in $\tau_{gas}$ due to the wavelengths considered, which apart from 940 nm (not included in

the AOD retrieval) have a negligible amount of water vapor absorption.

Langley regressions are found each day for two periods: morning and afternoon. The minimum in airmass (i.e.,

solar noon) separates each day's clear-sky times into morning and afternoon. The TOA DNI are then filtered by only

considering the interquartile range (i.e., 25[th]-75[th] percentile) to eliminate outliers and reduce noise (Koontz et al., 2013;

Ermold et al., 2013). The filtered TOA DNI are smoothed using a Savitzky-Golay filter. The filtered and smoothed TOA

DNI values for each wavelength considered are then utilized to retrieve the spectral HSR1 AOD for each clear-sky time.

The HSR1 AOD uncertainty is quantified. Since the HSR1 AOD is retrieved from Langley regressions, the AOD

uncertainty is independent of the HSR1 irradiance calibration. The HSR1 AOD uncertainties are due to: (1) uncertainties in

the TOA DNI, (2) cosine errors, and (3) dome lensing effects. The TOA DNI uncertainty is 1% as determined by the

standard error of the means. The cosine error uncertainty is 2% based on instrument design limits. The dome lensing effect

uncertainty is 1% as calculated from optical theory. The HSR1 AOD uncertainty is determined by considering the perturbation of the HSR1 AOD to the uncertainty sources. The resultant perturbation to the HSR1 AOD is ±0.02.

## 4 Results

Time series of the HSR1 $F_{total}$ and $F_{diffuse}$ at 500 nm are shown in Fig. 4 and of the HSR1 AOD at 500 nm are shown in Fig. 5. Time series of the MFRSR C1 $F_{total}$ and $F_{diffuse}$ at 500 nm as well as the CSPHOT AOD at 500 nm and MFRSR C1
AOD at 500 nm are also shown for comparison in Figs. 4 and 5, respectively. The comparison of these quantities is discussed in detail in the following subsections.

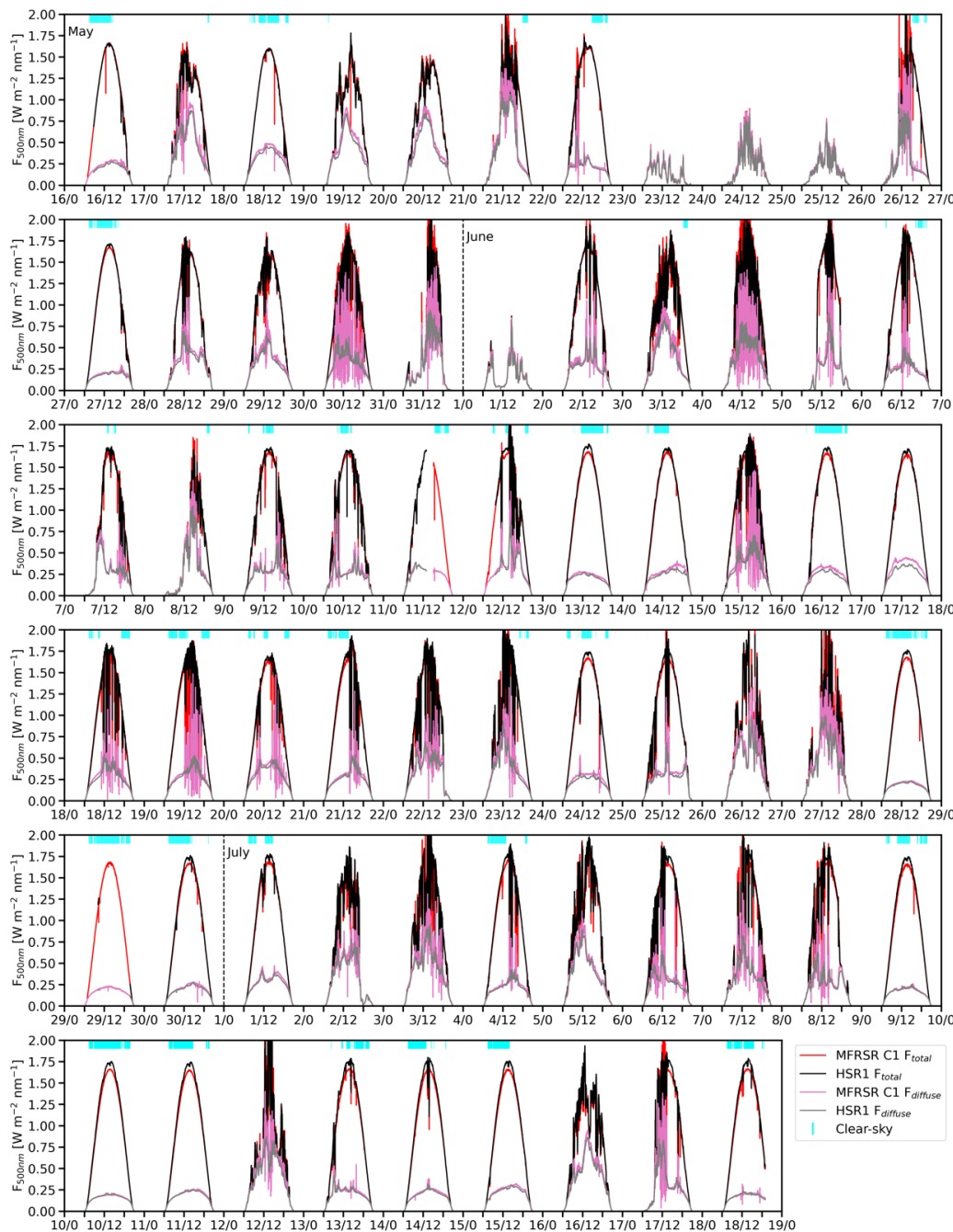

**Figure 4: Time series in LDT of the $F_{total}$ and $F_{diffuse}$ at 500 nm (W m$^{-2}$ nm$^{-1}$). The MFRSR C1 $F_{total}$ (red), HSR1 $F_{total}$ (black), MFRSR C1 $F_{diffuse}$ (pink), and HSR1 $F_{diffuse}$ (gray) are shown. The light blue vertical lines indicate clear-sky periods. The dashed vertical black lines indicate the start of each month and the x-axis tick marks indicate the day and hour.**

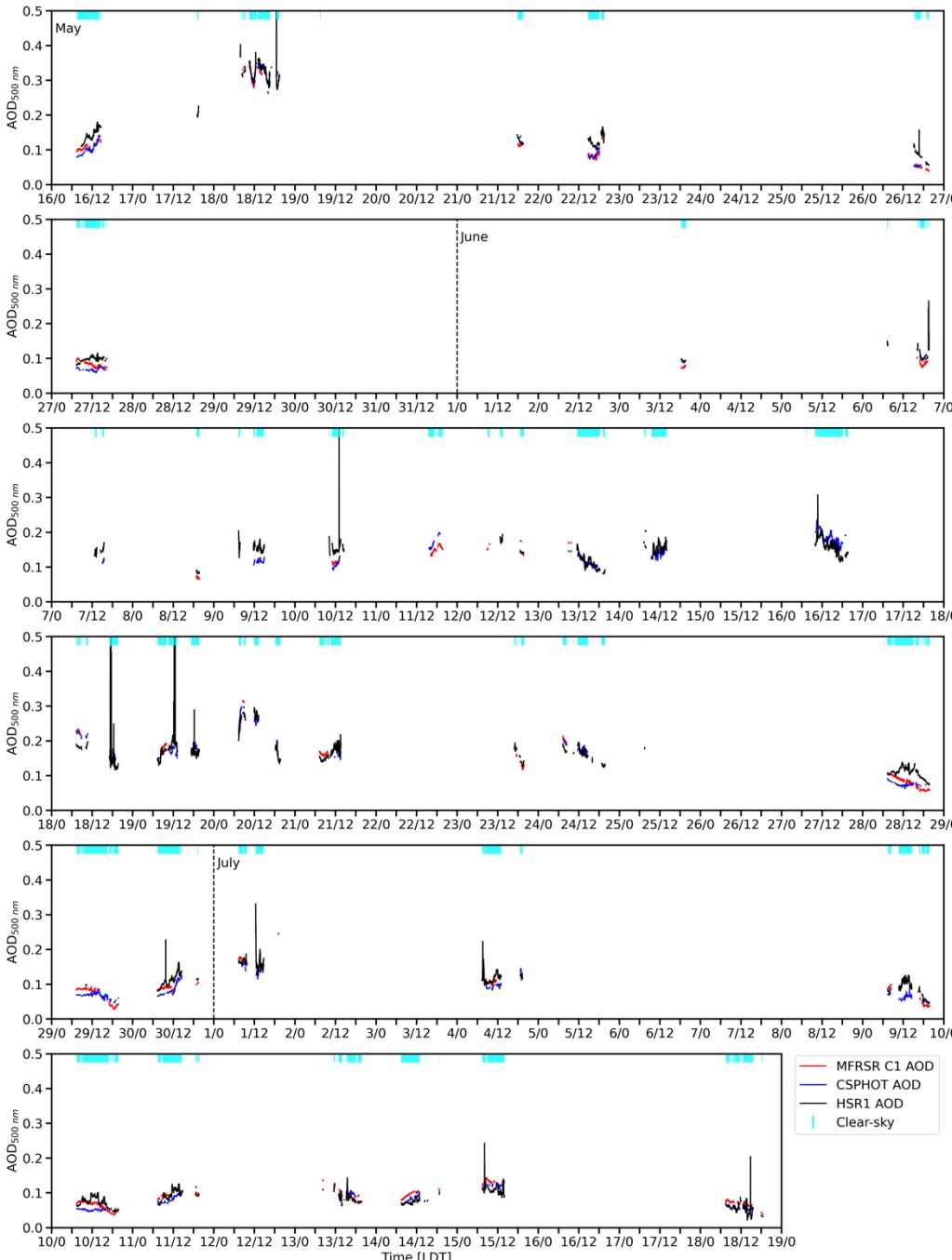

Figure 5: Time series (in LDT) of the AOD at 500 nm. The MFRSR C1 AOD (red), CSPHOT AOD (blue), and HSR1 AOD (black) are shown for clear-sky periods only. The light blue vertical lines indicate clear-sky periods. The dashed vertical black lines indicate the start of each month and the x-axis tick marks indicate the day and hour.

## 4.1 Irradiance comparison

The HSR1 $F_{total}$ and $F_{diffuse}$ were collocated and compared to those from the MFRSR C1 and MFRSR E13. The resultant comparison of the $F_{total}$ at 500 nm is shown in Fig. 6, $F_{diffuse}$ at 500 nm is shown in Fig. 7, and $F_{total}$ and $F_{diffuse}$ for all MFRSR wavelengths in Fig. 8. The MFRSR C1 and MFRSR E13 spectral irradiances are also compared to each other in Figs. 6-8 to provide context to the HSR1 comparison by showing the level of agreement between two instruments of the same model at the same location. In addition, the regression lines and the regression lines of the bias are shown in Figs. 6 and 7, which provides additional information on how the bias changes across different modes. The regression lines of the bias are constructed by regressing the bias (e.g., instrument 2 - instrument 1) with the reference instrument values (e.g., instrument 1).

For $F_{total}$, the mean (relative) differences at 500 nm for the HSR1 $F_{total}$ compared to the MFRSR C1 and MFRSR E13 $F_{total}$ are ~0.01 W m$^{-2}$ nm$^{-1}$ (1-2%) W m$^{-2}$ nm$^{-1}$. The comparison indicates that the HSR1 $F_{total}$ at 500 nm is slightly larger than those from both MFRSRs. However, the mean differences are relatively small demonstrating excellent agreement between the HSR1 $F_{total}$ and those from the two MFRSRs. For all MFRSR wavelengths, the relative ordering of the results are similar to those at 500 nm such that the mean HSR1 $F_{total}$ values are slightly larger than the mean $F_{total}$ values from the MFRSRs. The exception is at 415 nm where the HSR1 $F_{total}$ is slightly smaller than those from the MFRSRs by ~2-3%. The relative differences between the HSR1 $F_{total}$ and those from the MFRSRs are 8% or less except at 940 nm (~18%).

For $F_{diffuse}$, the mean (relative) differences at 500 nm for the HSR1 $F_{diffuse}$ compared to the MFRSR C1 and MFRSR E13 $F_{diffuse}$ are ~-0.035 W m$^{-2}$ nm$^{-1}$ (-10%) W m$^{-2}$ nm$^{-1}$. The HSR1 $F_{diffuse}$ values are smaller than those from both MFRSRs, which may be partially related to the underlying assumptions of isotropic diffuse radiation in the instrument design and a wider FOV than other instruments. Despite the consistently lower $F_{diffuse}$ values, the relative differences are within about 10%, which indicates good agreement between the HSR1 $F_{diffuse}$ and those from the MFRSRs. When considering all MFRSR wavelengths, the HSR1 $F_{diffuse}$ are smaller than those from both the MFRSRs by ~4-14%.

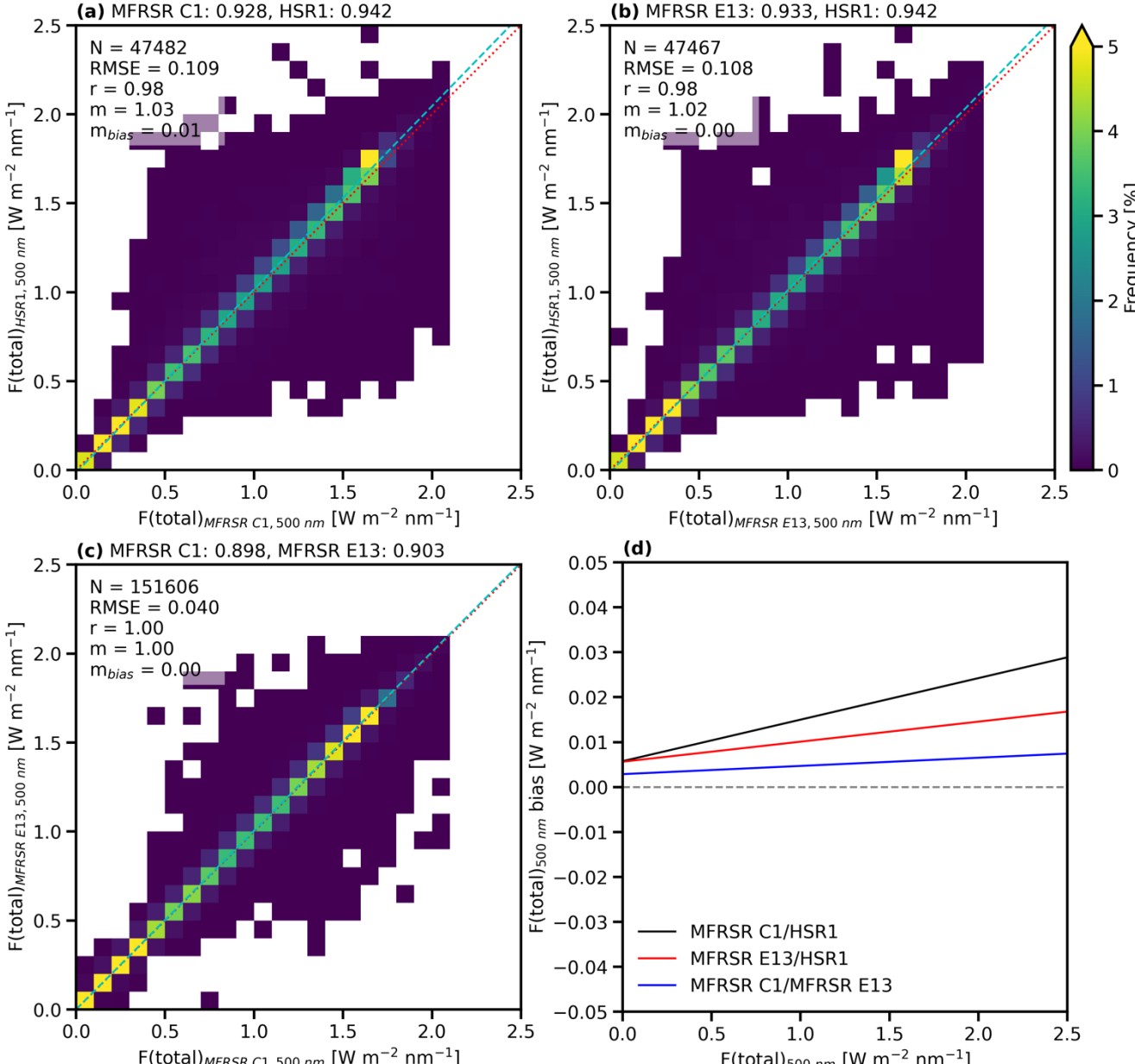

**Figure 6: Frequency histogram for $F_{total}$ at 500 nm (W m$^{-2}$ nm$^{-1}$) of collocated (a) MFRSR C1 and HSR1, (b) MFRSR E13 and HSR1, and (c) MFRSR C1and MFRSR E13. The mean values are given above each plot. The sample size (N), root mean squared error (RMSE), correlation coefficient (r), regression line slope (m), and bias regression line slope (m$_{bias}$) are shown in the top left of each plot. The 1:1 line is indicated by the dotted red line and the regression line is indicated by the dashed light blue line. (d) The regression lines of the bias are shown for MFRSR C1 and HSR1 (black), MFRSR E13 and HSR1 (red), and MFRSR C1 and MFRSR E13 (blue). The zero line is indicated by the dashed gray line.**

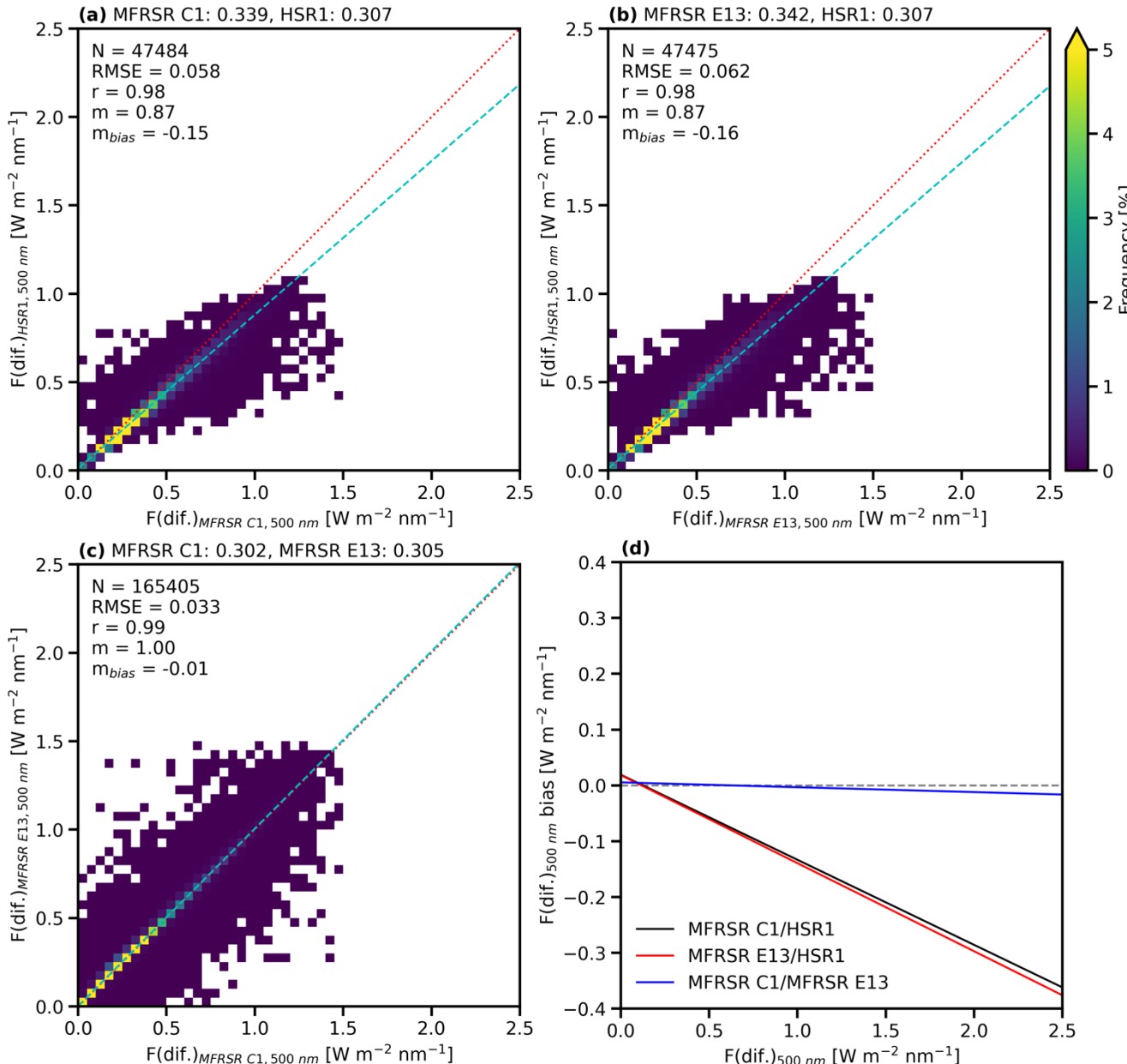

**Figure 7: The same as Figure 6 but for $F_{diffuse}$ at 500 nm.**

**4.1.1 Irradiance at 500 nm comparison**

For the HSR1 $F_{total}$ at 500 nm, the mean (relative) differences compared to the MFRSR C1 and MFRSR E13 are 0.014 W m$^{-2}$ nm$^{-1}$ (1.5%) and 0.010 W m$^{-2}$ nm$^{-1}$ (1.1%). In general, the HSR1 $F_{total}$ at 500 nm is slightly larger than those from both MFRSRs. However, the small mean differences, large correlation coefficients, regression slopes near 1, and bias regression slopes near 0 demonstrate excellent agreement between the HSR1 and the two MFRSRs in terms of the $F_{total}$ at

500 nm. Furthermore, the HSR1 $F_{total}$ at 500 nm is within the MFRSR uncertainty (3%; Table 1) of the MFRSR $F_{total}$ at 500 nm for 45.0% (MFRSR C1) and 54.8% (MFRSR E13) of the time.

For the HSR1 $F_{diffuse}$ at 500 nm, the mean (relative) differences compared to the MFRSR C1 and MFRSR E13 are -0.033 W m$^{-2}$ nm$^{-1}$ (-9.6%) and -0.036 W m$^{-2}$ nm$^{-1}$ (-10.4%). The $F_{diffuse}$ regression slopes are 0.87 between the HSR1 and both MFRSRs with negative bias regression slopes of -0.15 (MFRSR C1) and -0.16 (MFRSR E13), which is further indicative of the smaller HSR1 $F_{diffuse}$ compared to those from the MFRSRs. The relative differences are within about 10% indicating good agreement in a mean sense between the HSR1 and the MFRSRs in regards to the $F_{diffuse}$ at 500 nm. However, the HSR1 $F_{diffuse}$ at 500 nm is within the MFRSR uncertainty of the MFRSR $F_{diffuse}$ at 500 nm only 10.6% (MFRSR C1) and 7.3% (MFRSR E13) of the time.

The HSR1 $F_{diffuse}$ is smaller than those from both MFRSRs, which may be partially related to the instrument design in how the HSR1 measures the $F_{diffuse}$ as noted previously (Badosa et al., 2014). This includes the isotropic assumption and the HSR1 wider FOV than the other instruments. In reality, some of the forward-scattered circumsolar radiation is included in the HSR1 $F_{direct}$ which would be measured as $F_{diffuse}$ by instruments with a narrower FOV. This explains much of the underestimation of $F_{diffuse}$ observed in this comparison study.

The MFRSR C1 and MFRSR E13 were also compared in terms of the $F_{total}$ and $F_{diffuse}$ at 500 nm and found to agree well. The mean (relative) difference in the $F_{total}$ and $F_{diffuse}$ is 0.005 W m$^{-2}$ nm$^{-1}$ (0.5%) and 0.003 W m$^{-2}$ nm$^{-1}$ (0.9%), respectively. The $F_{total}$ and $F_{diffuse}$ correlation coefficients and regression slopes are near 1.00 with bias regression slopes of 0.00 to -0.01. The $F_{total}$ and $F_{diffuse}$ of the MFRSRs are within the MFRSR uncertainty of each other for 80.2% and 82.7% of the time, respectively. The comparison of the spectral irradiances between the two MFRSRs with the same instrument design and same data processing quantifies some of the uncertainty. It is also encouraging for the HSR1 that, with an independent instrument design and data processing, the HSR1 spectral irradiances agree well with those from the MFRSRs.

### 4.1.2 Irradiance at MFRSR wavelengths comparison

The HSR1 mean $F_{total}$ and $F_{diffuse}$ were compared to the mean $F_{total}$ and $F_{diffuse}$ from the MFRSRs for MFRSR wavelengths (i.e., 415, 500, 615, 673, 870, and 940 nm) in Fig. 8. For the $F_{total}$, the relative ordering of the comparison results are similar to those at 500 nm (Fig. 6): the mean HSR1 $F_{total}$ is slightly larger than those from the MFRSRs. The exception is at 415 nm where the HSR1 $F_{total}$ is slightly smaller than those from the MFRSRs by 1.9% (MFRSR C1) to 2.6% (MFRSR E13). The relative differences in $F_{total}$ between the HSR1 and the MFRSRs are 8% or less except at 940 nm. Similar to the comparison at 500 nm, the HSR1 $F_{total}$ at 415 nm was within the MFRSR uncertainty of the MFRSRs for 50% of the time. For 615-870 nm, the HSR1 $F_{total}$ was within the MFRSR uncertainty of the MFRSRs for 15-22% of the time. The HSR1 $F_{total}$ RMSE compared to the MFRSRs increases with wavelength until reaching the largest value at 615 nm and then the HSR1 $F_{total}$ RMSE decreases with wavelength.

At 940 nm, the HSR1 mean $F_{total}$ is larger than the MFRSRs by 21.0% (MFRSR C1) and 15.9% (MFRSR E13). Furthermore, the HSR1 $F_{total}$ was within the MFRSR uncertainty of the MFRSRs only 5-7% of the time. The larger relative

difference is partially due to the small magnitude of the mean $F_{total}$ at 940 nm of 0.098-0.119 W m$^{-2}$ nm$^{-1}$ noting that the

mean differences are 0.016-0.021 W m$^{-2}$ nm$^{-1}$. For reference, the MFRSR C1 and MFRSR E13 $F_{total}$ comparison at 940 nm is the largest relative difference spectrally as well with 4.5%, which is considerably higher than other wavelengths where the relative differences only ranged from 0.3 to 1.0%. This highlights the difficult and highly variable nature in measuring the $F_{total}$ at 940 nm where water vapor absorption is strong (Michalsky et al., 1995).

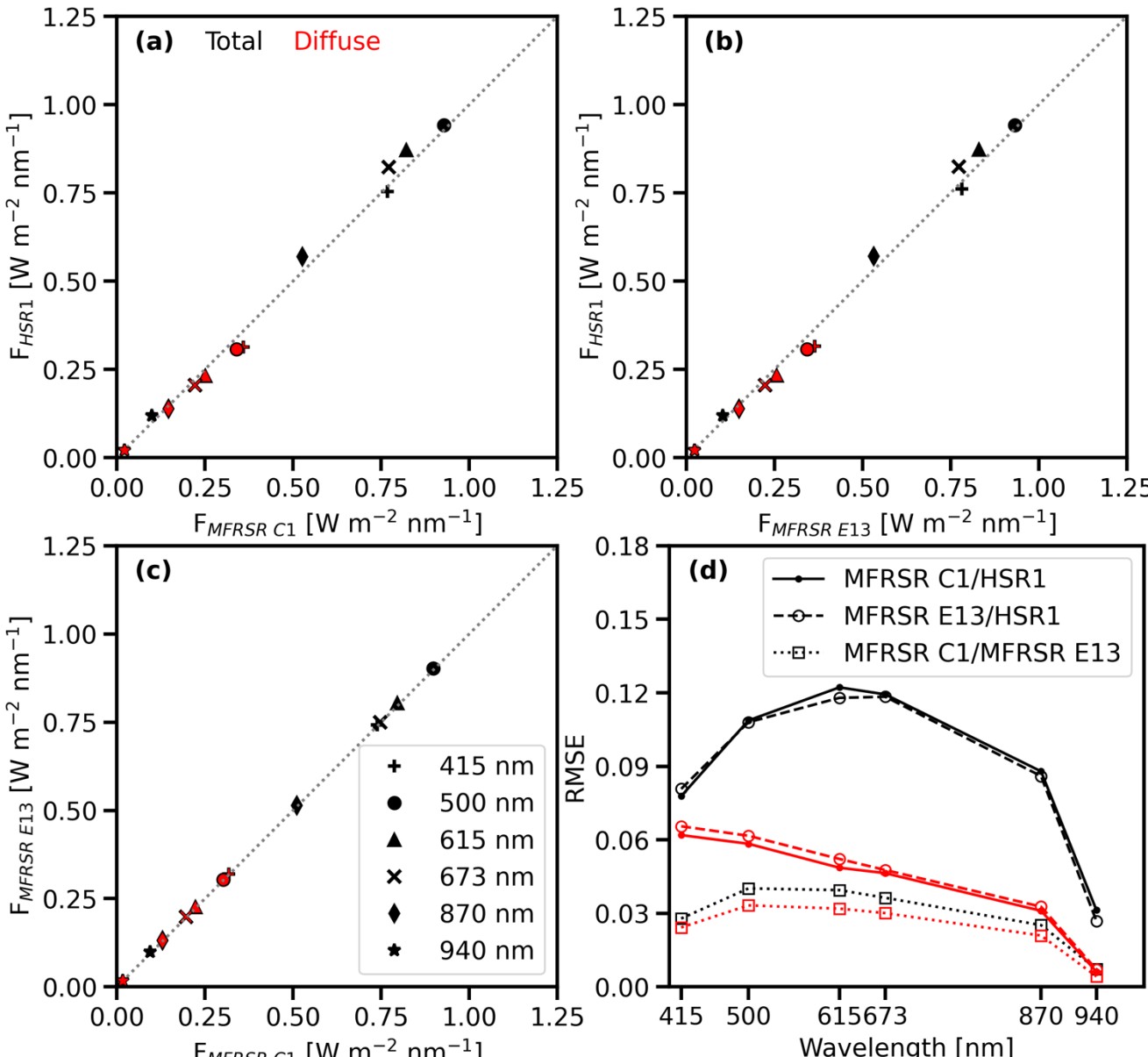

**Figure 8: Mean $F_{total}$ (black) and $F_{diffuse}$ (red) (W m$^{-2}$ nm$^{-1}$) of collocated (a) MFRSR C1 and HSR1, (b) MFRSR E13 and HSR1, and (c) MFRSR C1 and MFRSR E13. The wavelengths considered include 415 (plus sign), 500 (circle), 615 (triangle), 673 (x-mark), 870 (diamond), and 940 (star) nm. The 1:1 line is indicated by the dotted gray line. (d) Root mean square error (RMSE) are shown for $F_{total}$ (black) and $F_{diffuse}$ (red) for MFRSR C1 and HSR1 (solid line with dot), MFRSR E13 and HSR1 (dashed line with open circle), and MFRSR C1 and MFRSR E13 (dotted line with square).**

The $F_{diffuse}$ comparison at all MFRSR wavelengths follows a similar pattern in a relative difference sense to the comparison at 500 nm (Fig. 7): the HSR1 $F_{diffuse}$ are smaller than those from both the MFRSRs at all MFRSR wavelengths by ~4-14%. The relative differences range from -3.7% for the HSR1 $F_{diffuse}$ compared to those from the MFRSR C1 at 940 nm to -13.5% for the HSR1 $F_{diffuse}$ compared to those from the MFRSR E13 at 415 nm. The HSR1 $F_{diffuse}$ were within the MFRSR uncertainty only 2% of the time at 415 nm but 15-25% for 615-870 nm. The HSR1 $F_{diffuse}$ RMSE compared to the MFRSRs decreases with increasing wavelength, which is a similar spectral dependence to the $F_{diffuse}$ RMSE between the two MFRSRs.

Interestingly, the mean $F_{diffuse}$ for the HSR1 compared to those from the MFRSR C1 at 940 nm agree better than the MFRSR C1 and MFRSR E13 $F_{diffuse}$ at 940 nm of 9.8%. However, the mean differences for the $F_{diffuse}$ at 940 nm are small in magnitude at only 0.001-0.003 W m$^{-2}$ nm$^{-1}$. Similar to the $F_{total}$ comparison, the MFRSR C1 and MFRSR E13 $F_{diffuse}$ relative difference is largest at 940 nm compared to the relative differences at other MFRSR wavelengths. For context, the relative difference at 940 nm is nearly an order of magnitude larger than all other wavelengths (0.9-1.9%). In addition, the HSR1 $F_{diffuse}$ at 940 nm was within the MFRSR uncertainty of the MFRSRs by the same amount or more so (5-12%) than the MFRSRs were with each other (5.4%). This further highlights the challenges in measuring the spectral irradiance at 940 nm as two of the same instruments in the same location differ the most at this channel.

The impact of the MFRSR narrowband filter on the comparison results were quantified by considering the HSR1 spectra weighted by the MFRSR transmission spectra. In general, the HSR1 mean spectral irradiances decreased with the mean $F_{total}$ decreasing by 0.012 W m$^{-2}$ nm$^{-1}$ or less and the mean $F_{diffuse}$ decreasing by 0.005 W m$^{-2}$ nm$^{-1}$ or less. This resulted in the mean $F_{total}$ comparison between the HSR1 and the two MFRSRs improving by 0.1-1.0%, except at 940 nm where it improved by ~2.5% and at 415 nm where it worsened by 1.0-1.5%. For the mean $F_{diffuse}$, the comparison worsened by 1.5% or less except at 940 nm where it worsened by ~3.5%. Overall, the impact of the MFRSR narrowband filter is minimal on the results with changes in the HSR1 spectral irradiances and resultant comparison by ~0.01 W m$^{-2}$ nm$^{-1}$ (~1.5%) or less on average.

### 4.1.3 SASHe clear-sky irradiance comparison

The HSR1 $F_{total}$ and $F_{diffuse}$ were compared to the SASHe clear-sky irradiances. The SASHe clear-sky irradiances were also compared to the two MFRSRs. The resultant comparison for the $F_{total}$ and $F_{diffuse}$ are shown in Fig. 9.

Detailed comparisons of the $F_{total}$ at 500 nm (Fig. 9a) show the mean (relative) difference for the HSR1 $F_{total}$

compared to those from the SASHe is 0.017 W m$^{-2}$ nm$^{-1}$ (1.5%). The correlation coefficient is 0.98 with a regression slope of 1.06 and bias regression slope of 0.04, which further suggests that the HSR1 $F_{total}$ is larger than those from the SASHe. The SASHe $F_{total}$ were also compared to those from the two MFRSRs (Figs. 9b&c). The mean (relative) differences for the SASHe $F_{total}$ compared to those from the MFRSR C1 and MFRSR E13 are 0.004 W m$^{-2}$ nm$^{-1}$ (0.4%) and -0.001 W m$^{-2}$ nm$^{-1}$ (-0.1%). The regression slopes are less than 1 (0.97-0.98) and the bias regression slopes are negative (-0.04), which is due to

low-biased SASHe $F_{total}$ values at larger irradiances (>~1.5 W m$^{-2}$ nm$^{-1}$). The SASHe $F_{total}$ at 500 nm is within the MFRSR uncertainty of the MFRSR $F_{total}$ at 500 nm 72.2% (MFRSR C1) and 67.3% (MFRSR E13) of the time.

For the $F_{diffuse}$ at 500 nm, the mean (relative) difference for the HSR1 $F_{diffuse}$ compared to those from the SASHe is -0.019 W m$^{-2}$ nm$^{-1}$ (-6.4%). Similar to the HSR1 comparison in Fig. 7, the HSR1 $F_{diffuse}$ is smaller than that from the SASHe. The SASHe $F_{diffuse}$ is slightly smaller than those from the MFRSRs by -0.007 W m$^{-2}$ nm$^{-1}$ (-2.5%) and -0.011 W m$^{-2}$ nm$^{-1}$ (-

3.6%) for the MFRSR C1 and MFRSR E13, respectively. The SASHe $F_{diffuse}$ at 500 nm is within the MFRSR uncertainty of the MFRSR $F_{diffuse}$ at 500 nm 52.9% (MFRSR C1) and 44.2% (MFRSR E13) of the time.

A summary of comparisons of the mean $F_{total}$ and $F_{diffuse}$ irradiances at multiple wavelengths is shown between the SASHe and HSR1 (Fig 9i), MFRSR C1 (Fig 9j), and MFRSR E13 (Fig 9k). The SASHe and HSR1 mean $F_{total}$ comparisons are similar to those at 500 nm as the HSR1 $F_{total}$ is typically slightly larger than those from the SASHe by 0.08 W m$^{-2}$ nm$^{-1}$

(9%) or less except at 415 nm where the HSR1 $F_{total}$ is smaller by 0.01 W m$^{-2}$ nm$^{-1}$ (1.5%). The SASHe mean $F_{total}$ compared to those from the two MFRSRs agree within 2% or less for all wavelengths. The SASHe $F_{total}$ are within the MFRSR uncertainty of the MFRSRs for 48% of the time at 673 nm up to 76% at 415 nm. For the mean $F_{diffuse}$, the HSR1 $F_{diffuse}$ is smaller than those from the SASHe by 0.03 W m$^{-2}$ nm$^{-1}$ (9%) or less. The SASHe mean $F_{diffuse}$ compared to those from the MFRSRs are smaller by 0.02 W m$^{-2}$ nm$^{-1}$ or less. This corresponds to within 5% or less except at 673 and 870 nm where the

relative differences are within 5-8%. The SASHe $F_{diffuse}$ are within the MFRSR uncertainty of the MFRSRs from 19% of the time of the MFRSR E13 $F_{diffuse}$ at 673 nm up to 42% of the time of the MFRSR C1 $F_{diffuse}$ at 615 nm.

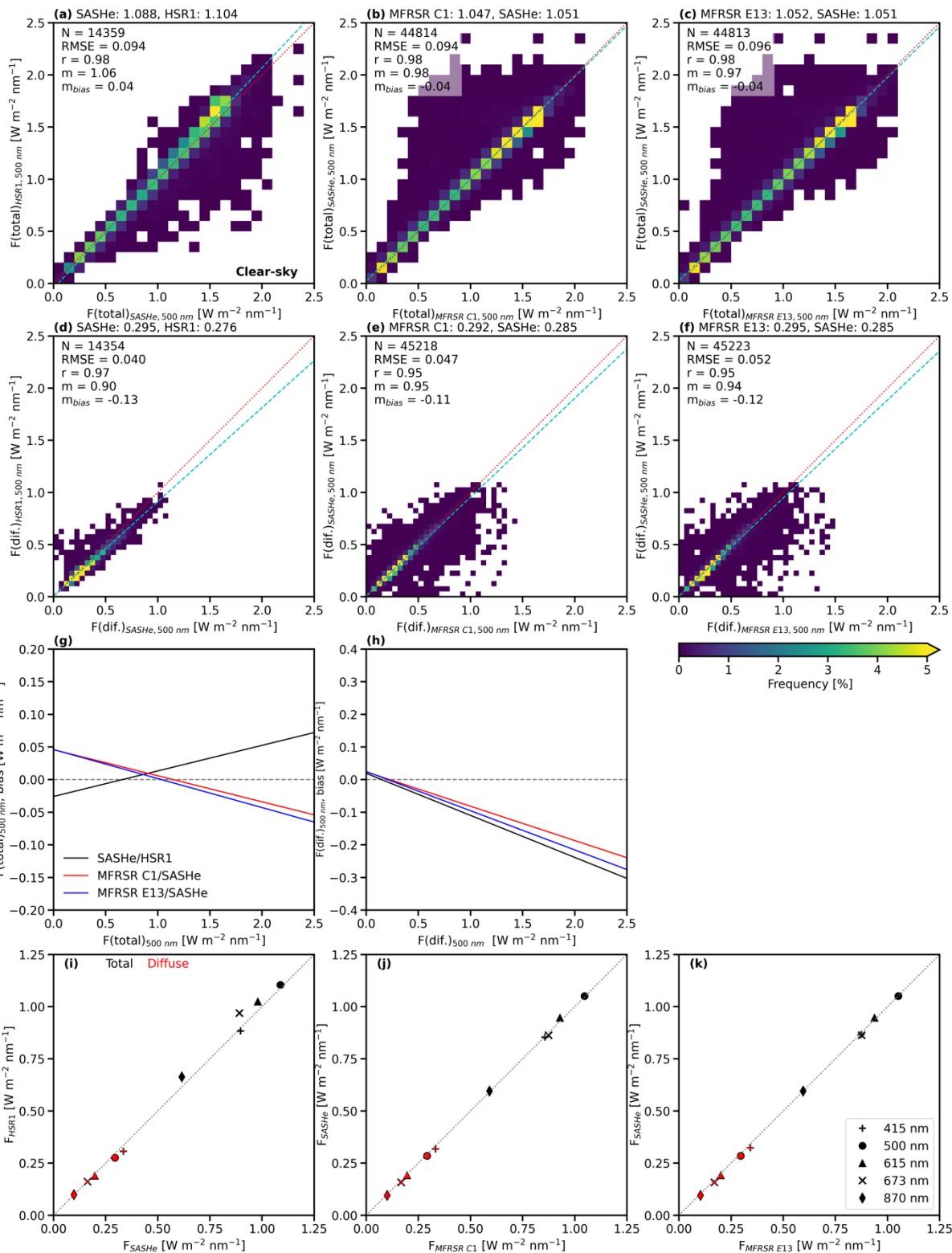

**Figure 9: Frequency histogram for the clear-sky (a-c) $F_{total}$ and (d-f) $F_{diffuse}$ at 500 nm (W m$^{-2}$ nm$^{-1}$) of collocated (left) SASHe and HSR1, (center) MFRSR C1 and SASHe, and (right) MFRSR E13 and SASHe. The mean values are given above each plot. The sample size (N), root mean square error (RMSE), correlation coefficient (r), regression line slope (m), and bias regression line slope ($m_{bias}$) are shown in the top left of each plot. The 1:1 line is indicated by the dotted red line and the regression line is indicated by the dashed light blue line. The regression lines of the (g) $F_{total}$ and (h) $F_{diffuse}$ bias are shown for SASHe and HSR1 (black), MFRSR C1 and SASHe (red), and MFRSR E13 and SASHe (blue). The zero line is indicated by the dashed gray line. (i-k) The mean clear-sky $F_{total}$ (black) and $F_{diffuse}$ (red) of collocated (left) SASHe and HSR1, (center) MFRSR C1 and SASHe, and (right) MFRSR E13 and SASHe at 415 (plus sign), 500 (circle), 615 (triangle), 673 (x-mark), and 870 (diamond) nm. The 1:1 line is indicated by the dotted gray line.**

## 4.2 AOD comparison

The HSR1 clear-sky AODs were compared to those from the CSPHOT, the MFRSRs, and the SASHe. The resultant comparison of the AODs at 500 nm is shown in Fig. 10 and for AODs at all overlapping wavelengths (i.e., 415, 440, 500, 615, 673, 675, and 870 nm) is shown in Fig. 11. The HSR1 AOD at 500 nm shows relative differences between 6 and 18% compared with retrievals from the other instruments. In general, the HSR1 AOD is larger than those from the other instruments except for the SASHe AOD. The mean differences in AOD are 0.01 or less with less than 10% relative differences (except for the MFRSR E13), which demonstrates excellent agreement. Furthermore, the mean difference between the CSPHOT AOD and HSR1 AOD is within CSPHOT's uncertainty of 0.01. For all overlapping wavelengths, better AOD agreement is found for the other instruments compared to each other than with the HSR1, which is similar to the spectral irradiance comparison.

### 4.2.1 AOD at 500 nm comparison

The HSR1 AOD at 500 nm shows mean (relative) differences with the CSPHOT, MFRSR C1, MFRSR E13, and SASHe of 0.010 (8.0%), 0.007 (6.4%), 0.017 (17.7%), and -0.008 (-6.2%), respectively. The correlation coefficients range from 0.91 to 0.94. The regression slopes are below 1.0 (0.88-0.95) except for the SASHe comparison (1.02). The regression slopes of the bias range from -0.08 to -0.18. With regression slopes less than 1 and negative slopes for the bias, this highlights that the HSR1 AOD is typically biased high at smaller AODs (~0.05-0.10) except for the SASHe AOD where the HSR1 AOD is biased low at larger AODs (~0.30-0.40). In addition, the HSR1 AOD at 500 nm is within the uncertainty of the CSPHOT and MFRSR AOD (0.01; Table 1) 28.1%, 32.5%, and 23.7% of the time for the CSPHOT, MFRSR C1, and MFRSR E13, respectively.

In general, the HSR1 AOD is larger than those from the other instruments except for those from the SASHe. Besides the MFRSR E13 AODs, the mean differences are 0.01 or less with less than 10% relative differences, demonstrating excellent agreement in the AOD at 500 nm between the HSR1 AOD and the CSPHOT, MFRSR C1, and SASHe AODs. In particular, the mean difference of 0.010 between the CSPHOT AOD and HSR1 AOD is encouraging, noting that the CSPHOT AOD uncertainty is 0.01 (Giles et al., 2019).

The CSPHOT, MFRSR C1, MFRSR E13, and SASHe AODs compare well with each other for AOD at 500 nm with relative agreements between 2 and 12%. This comparison provides context to the HSR1 AOD comparison by quantifying the level of agreement between established instruments and AOD retrievals. For the CSPHOT AOD comparison, the mean (relative) difference with the MFRSR C1, MFRSR E13, and SASHe AODs are 0.007 (7.5%), -0.004 (-4.4%), and 0.014 (10.0%), respectively. The mean (relative) difference in AODs between the MFRSR C1 and MFRSR E13, MFRSR C1 and SASHe, and MFRSR E13 and SASHe is -0.010 (-9.8%), 0.002 (2.3%), and 0.012 (12.0%), respectively. The correlation coefficients are also large, ranging from 0.94 to 0.99. The regression slopes are near 1 ranging from 0.90 to 1.00 except for the MFRSR AODs compared to the SASHe AODs which are 1.16 and 1.21. The regression slopes of the bias are negative with values of -0.11 or less, except for the MFRSR AODs compared to the SASHe AODs where the bias slope is positive with values of 0.14 and 0.18. In addition, the MFRSR AODs at 500 nm are within the uncertainty of the CSPHOT AOD for 22.7% and  78.2% of the time for the MFRSR C1 and MFRSR E13, respectively. The SASHe AODs at 500 nm are within the uncertainty of the CSPHOT and MFRSR AOD for 33.5%, 64.2%, and 34.0% of the time for the CSPHOT, MFRSR C1, and MFRSR E13, respectively.

Interestingly, the MFRSR AODs at 500 nm are within the uncertainty of each other only 18.2% of the time, indicating more agreement with the CSPHOT than retrievals from the same instrument type. Similar to the $F_{total}$ and $F_{diffuse}$ comparison, the MFRSR C1 and MFRSR E13 AOD comparison slightly disagree and that disagreement provides insight into part of the uncertainty of the measurement and AOD retrieval methods. The HSR1 AOD agrees with those from the other instruments, which is encouraging, indicating that some of the disagreement could be related to uncertainty inherent in the measurement and methods. The number of matching measurements for different instrument pairs are also quite variable, so the different comparisons may include different atmospheric conditions. This will also contribute to the variability between comparisons.

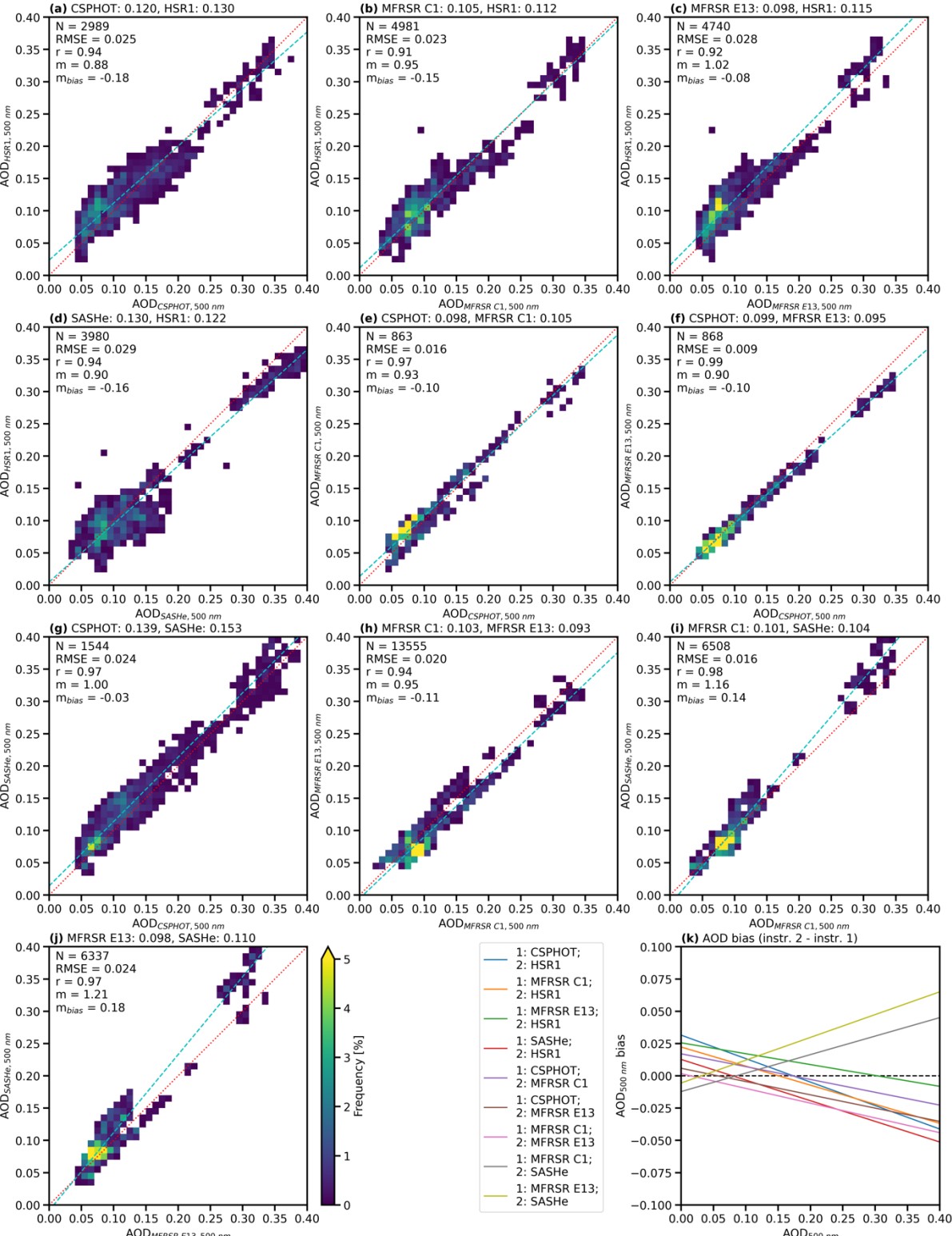

## 4.2.2 AOD at MFRSR wavelengths comparison

The HSR1 mean clear-sky AODs were compared to mean AODs from the CSPHOT, MFRSRs, and SASHe for overlapping wavelengths (i.e., 415, 440, 500, 615, 673, 675, and 870 nm) in Fig. 11. The relative ordering in the AOD comparison at all wavelengths is similar to those at 500 nm (Fig. 10): the mean HSR1 AOD is larger than those from the CSPHOT and the two MFRSRs except for the mean SASHe AOD, which is larger than the mean HSR1 AOD. The only spectral range where the HSR1 AOD is smaller than those from all other instruments is at 415 nm (MFRSRs and SASHe) and 440 nm (CSPHOT). The mean spectral HSR1 AOD for the 415 and 440 nm channels is smaller than those from the CSPHOT, MFRSR C1, MFRSR E13, and SASHe by 8.2%, 21.8%, 14.4%, and 23.1%. For 440 and 500 nm, the mean spectral HSR1 AOD comparison to the mean spectral CSPHOT AOD are within ~8% and ~0.01. However, the disagreement in the 675 and 870 nm AOD comparisons is larger: 0.021 (25.8%) and 0.030 (46.9%), respectively. For the MFRSR AODs, better agreement is found between the HSR1 and MFRSR C1 AODs than between the HSR1 and MFRSR E13 AODs. The relative differences between the mean spectral HSR1 AOD and MFRSR C1 AOD are 25% or less except for at 870 nm where it is 38.0%. In contrast, the relative differences between the mean spectral HSR1 AOD and MFRSR E13 AOD is 14-18% for smaller wavelengths (i.e., 415 and 500 nm) but 35-66% for larger wavelengths (i.e., 615, 673, and 870 nm). In addition, the HSR1 AOD RMSE compared to the AODs of the CSPHOT and the MFRSRs increases with increasing wavelength for the 500-870 nm spectral range. For the SASHe AOD, the HSR1 AOD is smaller by 10% or less except at 415 nm where the HSR1 AOD is smaller by 23.1% and 870 nm where the HSR1 is larger by 32.4%. Except for 415 nm, the HSR1 AOD RMSE compared to the SASHe AOD is nearly the same value spectrally (~0.03). The correlation coefficients are generally higher for smaller relative differences and are generally lower for larger relative differences in AOD (not shown). For example, the correlation coefficients are 0.91-0.95 for 415 and 500 nm but 0.61-0.86 for the 615, 673, and 870 nm HSR1 AOD comparisons. Similarly, the regression slopes are closer to 1 and the regression slopes of the bias are closer to 0 for smaller relative differences and the opposite is seen for larger relative differences. For example, the regression slopes range from 0.65 to 1.22 for the 870 nm HSR1 AOD comparisons. Furthermore, the bias regression slopes are within ~0.2-0.3 except for 870 nm where the slopes range from -0.49 to -0.84. HSR1 AODs are less frequently within the uncertainty of the CSPHOT and MFRSR AODs for shorter and longer wavelengths compared to 500 nm. However, the value is consistent across the comparison ranging from 15-25% except for the HSR1 AODs within the MFRSR C1 at longer wavelengths (i.e., 615, 673, and 870 nm) where the value is 28-39% of the time.

The mean CSPHOT AOD were compared to the two MFRSRs and SASHe AODs at the two overlapping wavelengths (i.e., 500 and 870 nm) and the mean AODs are largely found to agree well. The AOD comparison at 500 nm is described above. For 870 nm, the mean (relative) difference between the CSPHOT AOD with those from the MFRSR C1, MFRSR E13, and SASHe is 0.009 (18.3%), -0.002 (-3.5%), and 0.04 (6.0%), respectively. The correlation coefficients are also large between the CSPHOT AOD and those from the other instruments ranging from 0.87 to 0.98. The regression slopes are slightly further from 1 and the bias regression slopes are more negative for the 870 nm AOD comparison, noting that the regression slope range is 0.85 to 1.03 and the bias regression slope range is -0.14 to -0.16. In addition, the AODs at 870 nm are within the uncertainty of the CSPHOT AOD for 25.0%, 89.0%, and 32.0% of the time for the MFRSR C1, MFRSR E13, and SASHe, respectively. Interestingly, the MFRSR E13 AOD at 870 nm is again within the uncertainty of the CSPHOT AOD at 870 nm more often than the MFRSR C1 AOD at 870 nm (26.8%).

The AODs from the two MFRSRs are compared to each other as well. The mean differences are all 0.009-0.011 with relative differences of 8-17%. The relative frequency of the MFRSR AODs are within the uncertainty of each other is 30% at 415 nm and lower at higher wavelengths of 615-870 nm (14-27%). The MFRSRs and the SASHe AODs were also compared to each other. The SASHe AOD is typically larger than those from the two MFRSRs ranging from a relative difference of 2% to up to 41% at 615 nm. The exception is when the SASHe AOD is smaller than the MFRSR AOD which includes at 940 nm and for the MFRSR C1 AOD at 415 nm. The SASHe AODs are within the uncertainty of the MFRSR AODs for 21% of the time at 615 nm for the MFRSR E13 and up to 49% at 415 nm for the MFRSR E13. The AOD RMSE between the other instruments is the same or smaller in value than the HSR1 AOD RMSE, which can be seen by comparing Fig. 11i to Fig. 11j. In general, better agreement is found between AODs derived from the other instruments than with the HSR1 AOD, particularly at larger wavelengths.

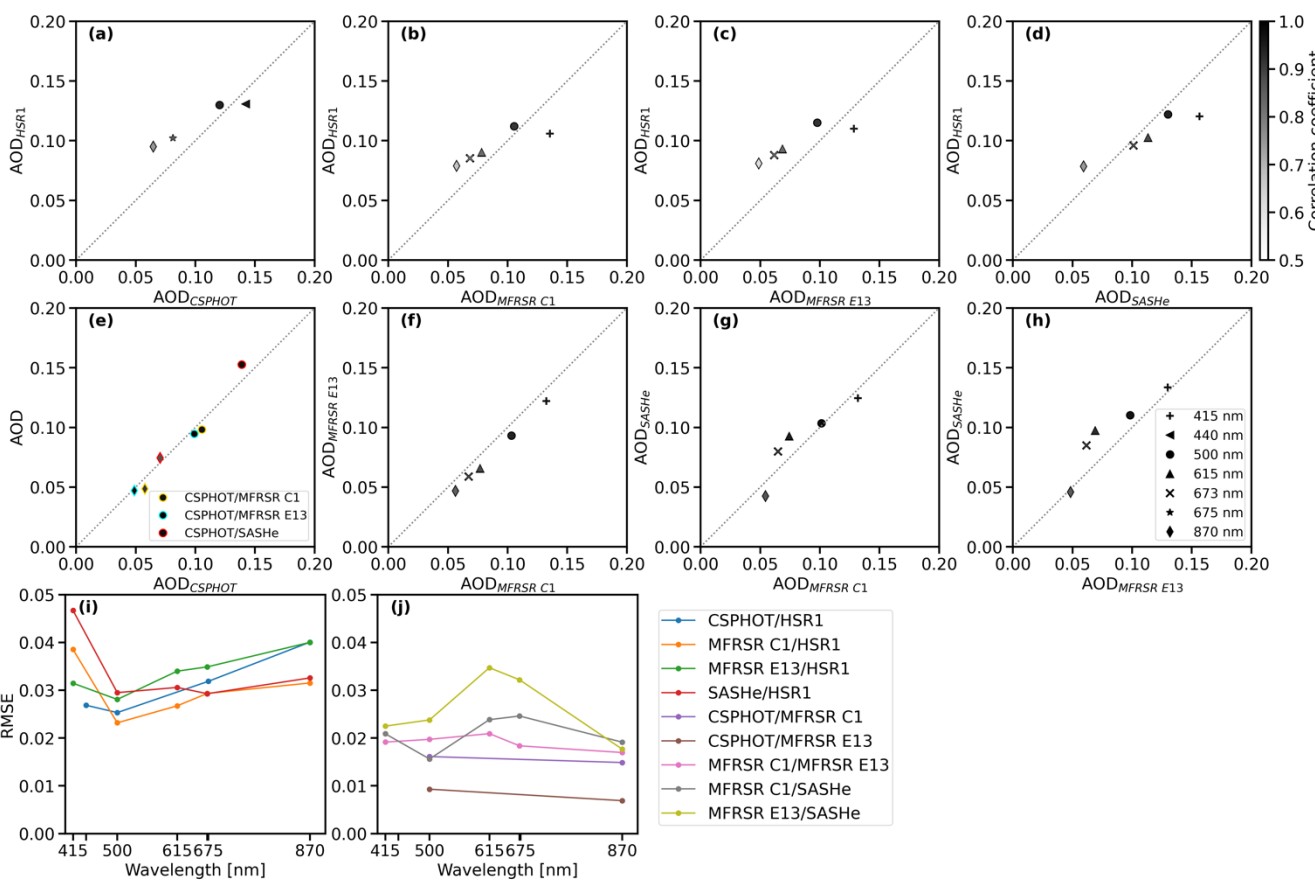

**Figure 11: Mean AOD (black) of collocated (a) CSPHOT and HSR1, (b) MFRSR C1 and HSR1, (c) MFRSR E13 and HSR1, (d) SASHe and HSR1, (f) MFRSR C1 and MFRSR E13, (g) MFRSR C1 and SASHe, and (h) MFRSR E13 and SASHe. The shading indicates the correlation coefficient. Mean AOD of collocated CSPHOT with MFRSR C1 (yellow outline), MFRSR E13 (light blue outline), and SASHe (red outline) are shown in (e). The wavelengths considered include 415 (plus sign), 440 (left pointing triangle), 500 (circle), 615 (triangle), 673 (x-mark), 675 (star), and 870 (diamond) nm. The 1:1 line is indicated by the dotted gray line. Root mean square error (RMSE) are shown for (i) HSR1 AOD and other instruments, and (j) other instruments between each other.**

## 4.3 Diffuse ratio comparison

The HSR1 diffuse ratios were compared to the spectral diffuse ratios from the MFRSRs and SASHe and to the broadband diffuse ratios from RADFLUX. This gives an irradiance comparison that is not dependent on the instrument calibration. It is also a useful quantity to look at the impact of clouds on the irradiance.

For the spectral comparison, the mean (relative) diffuse ratio differences are typically ~-0.05 or less (12% or less) for the MFRSR wavelengths except for the MFRSR diffuse ratio comparison at 940 nm where the relative difference is ~17-20%.

For the broadband comparison, the HSR1 mean integrated diffuse ratio is typically smaller than the broadband diffuse ratios by 0.04 (9%). The similarity between the broadband and spectral diffuse ratio comparison suggests that the underestimation in the HSR1 $F_{diffuse}$ measurements is the likely source of the broadband disagreement more so than the HSR1 measuring a portion of the solar spectrum.

### 4.3.1 Spectral diffuse ratio comparison

As expected, the spectral diffuse ratio comparison reflects the fact that $F_{total}$ has better agreement between instruments than $F_{diffuse}$. Overall, the HSR1 spectral diffuse ratio is typically smaller than those from the two MFRSRs at 500 nm although general agreement is found with relative differences of 8% (Figs. 12a&b). Similar to the $F_{diffuse}$ comparison at 500 nm (Fig. 7), the mean HSR1 spectral diffuse ratio is smaller than those from the two MFRSRs at all other MFRSR wavelengths as well (not shown). For all times, the mean HSR1 spectral diffuse ratio is smaller than those from both MFRSRs by 0.05 (10%) or less at all wavelengths except for 940 nm where it is 0.06-0.08 (17-20%). The results are similar when considering clear-sky times: HSR1 diffuse ratios are smaller than MFRSRs by 0.05 or less with relative differences of 12% or less except at 940 nm where the relative difference is 32-42%.

The SASHe clear-sky spectral diffuse ratios were also compared at 415, 615, 673, and 870 nm. The relative differences at other wavelengths are found to be similar to the 500 nm relative differences. The HSR1 diffuse ratio is smaller than the SASHe diffuse ratio by 6% or less except at 870 nm where it is larger by 3% whereas the SASHe diffuse ratio is smaller than those from the two MFRSRs by 2 to 12%. The diffuse ratios from the two MFRSRs were also compared to each other, with the 5th to 95th percentile ranging from -5% to 12%, except for 940 nm where the range is -2% to 37%.

The high correlation (0.96-0.98) at 500 nm between the HSR1 and various instrument pairs is shown in Figs. 12a-c. This is similar or slightly higher than the correlation between the MFRSRs and the SASHE (Figs. 12e&f), though does not match the near perfect correlation between the two MFRSRs (Fig. 12d).

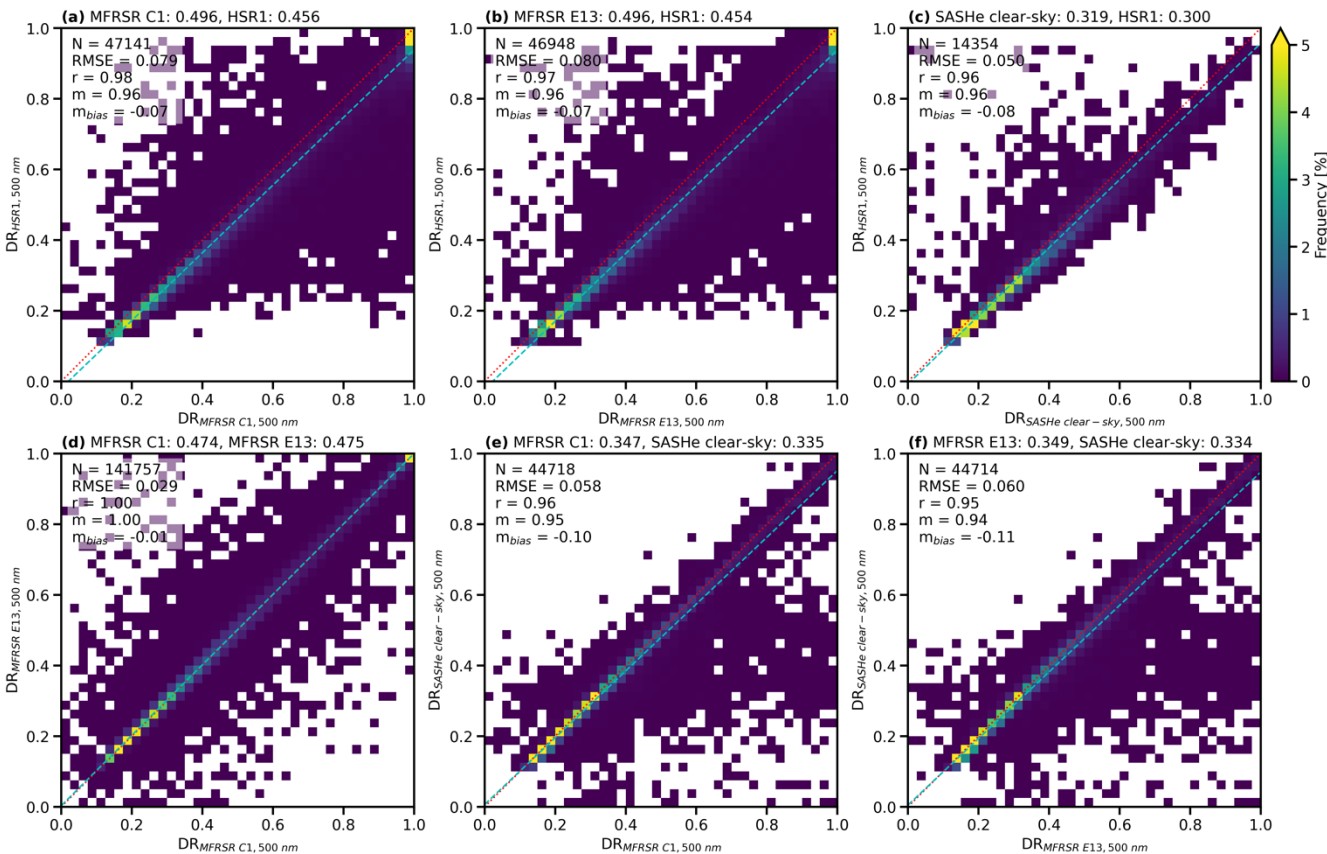

**Figure 12: Frequency histogram for the diffuse ratio at 500 nm of collocated (a) MFRSR C1 and HSR1, (b) MFRSR E13 and HSR1, (c) SASHe and HSR1, (d) MFRSR C1 and MFRSR E13, (e) MFRSR C1 and SASHe, and (f) MFRSR E13 and SASHe. The mean values are given above each plot. The sample size (N), root mean square error (RMSE), correlation coefficient (r), regression line slope (m), and bias regression line slope ($m_{bias}$) are shown in the top left of each plot. The 1:1 line is indicated by the dotted red line and the regression line is indicated by the dashed light blue line. Note that SASHe diffuse ratios are limited to clear-sky conditions.**

### 4.3.2 Broadband diffuse ratio comparison

The HSR1 integrated diffuse ratio is constructed by considering the $F_{diffuse}$ and the $F_{total}$ both integrated from 400 to 1000 nm and then dividing the integrated $F_{diffuse}$ by the integrated $F_{total}$. The HSR1 integrated diffuse ratios were compared to the broadband diffuse ratios from RADFLUX (Sect. 2.2.4). The motivation of this comparison is to understand if the HSR1 integrated diffuse ratio captures the diffuse ratio in the absence of a diffuse solar broadband irradiance observation (e.g., only total broadband SW measurements) despite measuring only a portion of the solar spectral range.

The resultant diffuse ratio comparison is shown in Fig. 13. The HSR1 integrated diffuse ratio is found to typically be smaller than the broadband diffuse ratios. In terms of the mean diffuse ratio, the HSR1 diffuse ratio is smaller than the broadband diffuse ratio by 0.036 (8.5%) for all times (Fig. 13a) and 0.014 (7.8%) for clear-sky times (Fig. 13b). The diffuse

ratio comparison is also separated into overcast and partial cloudy-skies (not shown) and the mean (relative) differences are 0.047 (-5.0%) and 0.043 (-11.6%), respectively. In general, the HSR1 integrated diffuse ratio is 12% smaller or less with closer agreement for clear-sky in absolute difference and overcast conditions in relative difference and worse agreement during the dominant mode of partial cloudy-skies, which accounts for ~60% of all times.

To gauge the impact of the diffuse ratio error in terms of the irradiance errors, the error in the broadband diffuse

irradiance ($F_{broadband,diffuse}$) is considered by comparing the broadband total irradiance ($F_{broadband,total}$) and HSR1 integrated diffuse ratio ($DR_{HSR1}$) to the $F_{broadband,diffuse}$:

$$F_{diffuse,error} = F_{broadband,total} \; x \; DR_{HSR1} - F_{broadband,diffuse} \;. \tag{6}$$

The relative percent difference is shown in Fig. 13c and the resultant irradiance error is shown in Fig. 13d. The mean $F_{diffuse,error}$ is -16.7 and -7.9 W m$^{-2}$ for all times and clear-sky times, respectively. The measurement uncertainty of the

570 $F_{broadband,diffuse}$ is ±3% (Sect. 2.2.4). If the $F_{broadband,diffuse}$ is determined by the $DR_{HSR1}$, then the $F_{diffuse,error}$ considering the $DR_{HSR1}$ are within the $F_{broadband,diffuse}$ uncertainty only 16.5% (all times) and 18.3% (clear-sky times) of the time.

Interestingly, the broadband diffuse ratio comparison results are similar to those from the spectral diffuse ratio comparison (Fig. 12). This suggests that the biased low HSR1 $F_{diffuse}$ measurements due to the instrument design may be the

575 dominant feature that explains the difference in the broadband diffuse ratio and not that the HSR1 measures less of the solar spectrum than a broadband radiometer. Furthermore, the smaller solar spectral range of the HSR1 would induce a high bias as the diffuse ratio decreases with increasing wavelength. This further suggests that the low bias in the HSR1 diffuse measurements is the dominant feature for the low diffuse ratio bias.

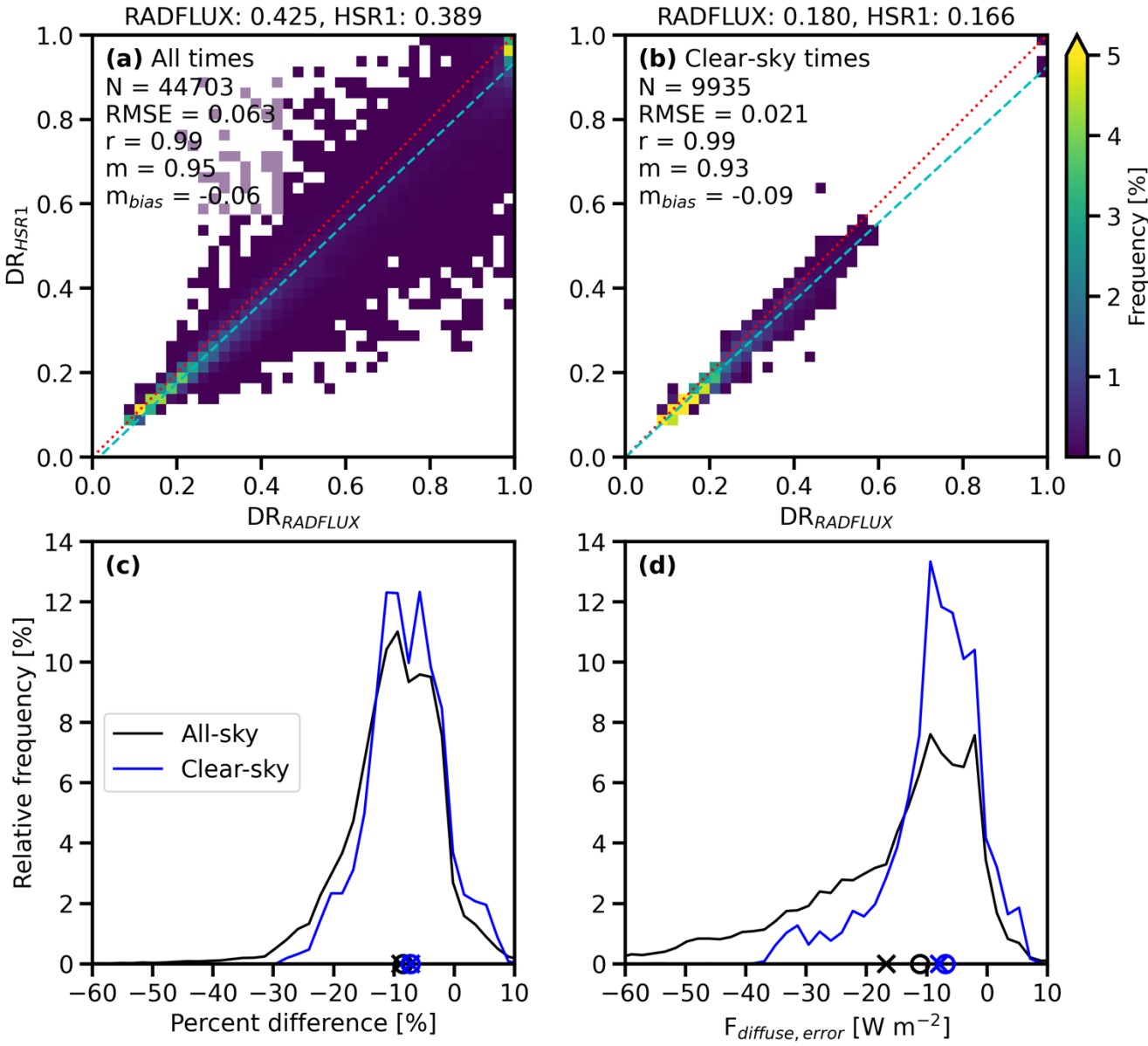

Figure 13: Frequency histogram for collocated diffuse ratio (DR) between RADFLUX and HSR1 for (a) all times, and (b) clear-sky times. The mean values are given above each plot and the sample size (N), root mean square error (RMSE), correlation coefficient (r), slope of the regression line (m), and slope of the regression line of the bias ($m_{bias}$) are shown in the top left of each plot. The 1:1 585 line is indicated by the dotted red line and the regression line is indicated by the dashed light blue line. Relative frequency plots of the (c) diffuse ratio percent difference between RADFLUX and HSR1 and (d) irradiance error in the broadband diffuse irradiance due to the HSR1 measured diffuse ratio ($F_{diffuse,error}$). The relative frequencies for all-sky times are in black and for clear-sky times are in blue. The mean value is denoted by a x-mark and the median is denoted by the open circle along the x-axis.

## 5 Discussion

In this study, the HSR1 is evaluated for future use as a hyperspectral radiometer. As shown in Figs. 6, 7, and 10, the HSR1 shows a close agreement with both the MFRSR $F_{total}$ and $F_{diffuse}$ at 500 nm, and the CSPHOT AOD at 500 nm. This is encouraging, and indicates that the HSR1 can give comparable results to these instruments at modest cost, or in situations where the current instruments are difficult to operate, e.g., remote sites, or moving platforms such as boats or planes. The ability of the HSR1 to give continuous measurements, both in time and spectrally, may also open up new opportunities.

### 595    5.1 Total irradiance measurements

         As shown in the selected spectra in Fig. 2, and the summary comparisons in Figs. 8 and 11, the HSR1 spectral values are generally in good agreement at 415 nm and 500 nm, with the HSR1 measuring higher values at higher wavelengths. This pattern is in agreement with the extra-terrestrial values calculated by the Langley process (see later discussion).

The HSR1 continuous spectral measurements (as with the SASHe) can also be used to match specific spectral sensitivities, such as photosynthetically active radiation (PAR) for agricultural research, photopic eye-response (illuminance) for architectural use, or photovoltaic (PV) panel sensitivities for PV research. An example comparison of the HSR1 with a Kipp & Zonen PAR sensor is shown in Appendix B.

### 5.2 Diffuse irradiance measurements

A distinctive feature of the comparisons in Fig. 4 is that the $F_{diffuse}$ by the MFRSR is noticeably more variable in broken cloud conditions than the HSR1 measurement. This variation may be due to several possibilities:

1. The HSR1 measures both $F_{total}$ and $F_{diffuse}$ at the same time, whereas the MFRSR measures these sequentially during a 20 s scan of the shadowband.
2. The HSR1 measurements are averaged over a 1-min period with a 10 s sampling interval, whereas the MFRSR
measurements are the 20 s closest to the HSR1 time. Fast moving clouds can change the irradiance rapidly in these conditions.
3. It is possible that the various logger clocks are not always accurately aligned.

These differences in measurement and time synchronisation will also explain the low-frequency background scatter of points in the irradiance comparison plots (Figs. 6 and 7).

The other distinctive feature is the low $F_{diffuse}$ measurement of the HSR1 relative to all the reference instruments. This was also noted by Badosa et al. (2014), and is a feature of the shading mask design. This low bias in $F_{diffuse}$ has several possible causes:

1. The wide FOV of the HSR1 optics compared to the narrower FOV of the MFRSR, which means that forward-scattered circumsolar radiation is excluded from the HSR1 $F_{diffuse}$ measurement, but included in the MFRSR

measurement, which is able to measure the circumsolar component directly. Interestingly, the SASHe appears to show some similarities to the HSR1 in this regard. The circumsolar fraction increases with increasing AOD and cloud optical depth (COD), and hence, $F_{diffuse}$. Both SASHe and HSR1 show a reducing diffuse ratio with increasing diffuse irradiance, implying more of the circumsolar irradiance is included in $F_{direct}$ compared to the other references.

2.  Manufacturing tolerances within the HSR1 shading mask may deviate from the assumption that the open areas are exactly 50% of the full hemisphere.

## 5.3 AOD measurements

The HSR1 AOD calculation is based on the Langley method (Sect. 3), so it is independent of the HSR1 calibration accuracy, and provides an independent check on the HSR1 calibration across those wavelengths where the Langley method

applies.

At 500 nm, the two MFRSRs agree closely with each other, and with the CSPHOT. HSR1 and SASHe are a little more variable between each other and CSPHOT. The HSR1 RMSE compared to CSPHOT is typically up to twice that of the MFRSRs, and similar to SASHe. This pattern is also shown in the RMSEs at other wavelengths (Figs. 11i&j).

The HSR1 AOD at 500 nm also shows slope less than unity against CSPHOT (Fig. 10a), as seen in previous

comparisons (Wood et al., 2017). In the previous study, correlation with the CSPHOT was improved by an empirical correction. This has not been applied here, but further analysis will be presented in a future paper.

We also note that both HSR1 and SASHe can both generate spectrally continuous AOD measurements, though these are not shown here. These may enable distinguishing between coarse and fine aerosols, or cloud contamination, as suggested in Norgren et al. (2022).

## 5.4 Calibration

Calibration against a standard lamp provides a good starting calibration, but there may be improvements possible. The generally low light levels from the FEL lamp (~0.07 W m$^{-2}$ nm$^{-1}$) can be difficult to scale up to sunshine outdoors (~2 W m$^{-2}$ nm$^{-1}$) without introducing errors, which can affect the accuracy of measurement outdoors. They do, however, give a very smooth stable calibration over the whole spectral range.

The Langley method provides a comparison with the solar extra-terrestrial (ET) spectrum in the wavelength ranges that are unaffected by gas absorption bands. As the HSR1 outputs are calibrated in W m$^{-2}$ nm$^{-1}$, the Langley intercept values should be the same as the known solar ET.

Figure 14 shows the solar ET spectrum from SMARTS2 v2.95 smoothed to 3 nm bandwidth to match the HSR1, and the median of the Langley intercept values based on the HSR1 as originally calibrated. This shows a deviation similar to

that shown in Fig. 2. Note also that this shows an HSR1 wavelength calibration offset of ~5 nm.

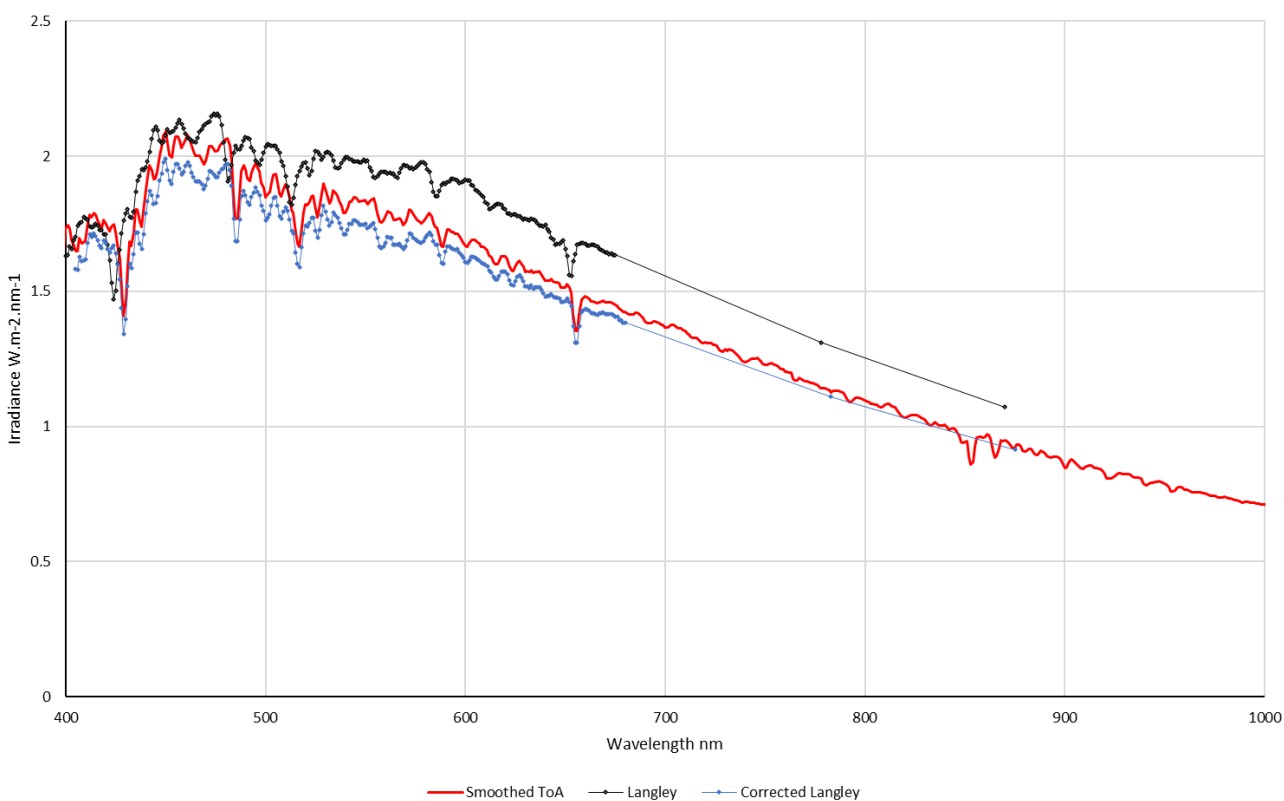

**Figure 14: Extra-terrestrial spectral irradiance from SMARTS2 (red), the Langley intercept values as calculated from the original HSR1 measurements (black), and the Langley intercept values adjusted according to the post-deployment calibration check with wavelengths adjusted to match the SMARTS2 spectral features (blue).**

The Langley intercept corrected according to the post-deployment calibration check, and with the wavelength calibration offset applied, is also shown. This is in much better agreement with the Solar ET – the RMSE between the Langley intercepts and the Solar ET has halved from 0.16 to 0.08. This method may enable a continuing check on calibration during operation, as long as there are sufficient clear-sky periods to give a robust Langley calculation.

### 5.5 Future work

We have identified several areas for more detailed study, which we would hope to present in a later publication. These are described briefly here.

For AOD retrievals, the use of the full spectral range of the HSR1 may enable better AOD retrievals, in particular
using the slope and spectral shape of the calculated optical depth from the HSR1 to determine the presence and quantity of

light cloud in apparently clear skies (see Fig. 1 of Norgren et al. 2022 and accompanying description). It may also be possible to improve the HSR1 AOD calculations by applying a correction for the wider FOV, as suggested in Wood et al. (2017), but with a better theoretical basis, as briefly described in Appendix A of Norgren et al. (2022). We would also like to explore the use of the HSR1 spectra for retrievals of other quantities such as water vapor or ozone.

In the area of instrument calibration, there are potential improvements to be made over the standard lamp calibration, in using the Langley technique to correct or monitor the instrument calibration over time. The reasons for the low diffuse sensitivity should also be investigated and corrected where possible. The effects of correcting for the dome lensing variability first noted in Badosa et al. (2014) will also be investigated further, and may reduce some of the variabilities in the $F_{total}$ and $F_{diffuse}$, and retrieved AOD. Initial analysis indicates that the dome lensing effect on the results in
this study are small with a change of 0.01 or less in the $F_{total}$, $F_{diffuse}$, and AOD at 500 nm.

     Other future instrument designs plan to address the measurement noise at the lower (below 400 nm) and upper (above 950 nm) wavelengths. The HSR1 demonstrated the capability to measure $F_{total}$ and $F_{diffuse}$ at wavelengths outside the spectral range focused on in this study of 400-950 nm (Figs. 2b&c). Future instrument designs plan to overcome the current prototype's noise and the extended spectral range may be a high-quality measurable quantity in the future.

**6 Conclusion**

     A new hyperspectral radiometer called the HSR1 was evaluated in terms of operability and performance in measuring surface irradiances and aerosol optical properties. This new instrument provides several distinct advantages and disadvantages compared to other instrumentation available for measuring spectral irradiances and AOD. The fixed-shading pattern that requires no moving parts makes this instrument unique among the instruments compared in this study. All other
instrumentation required alignment with the sun, which requires sun tracking and ultimately limits the ability of the instrumentation to operate in remote environments or on moving platforms (e.g., ships and aircraft). The trade-off, however, is that the wider FOV from this shading mask leads to inclusion of more of the circumsolar scattering in the direct rather than diffuse irradiance, and a corresponding underestimation in diffuse irradiance, that is wavelength dependent. The evaluation analysis indicates that the mean AOD retrieved from the new hyperspectral radiometer is typically within uncertainty limits
(0.01) of existing filter-based instruments including a CSPHOT and two MFRSRs. There is, however, more wavelength-dependent systematic disagreement in AOD retrievals from spectrometer-based instruments and filter-based instruments, than there is between different filter-based instruments. While spectrometers give unique and valuable information in the spectral dimension of the measurement, the lower signal to noise ratio in the measurements along with increased challenges from straylight detection at shorter and longer wavelengths lead to higher uncertainty in retrieved AODs than in filter-based
instruments.

     The analysis was limited to just irradiance and AOD comparisons in this study due to the number of comparison data sources available, although retrievals of other atmospheric and land surface properties are possible with hyperspectral

measurements. The scientific need for hyperspectral radiometers will continue to increase in importance in the future as weather, climate, and renewable energy forecasting advance to incorporate spectral characteristics of aerosols and clouds.

With the advancement of hyperspectral radiometers to meet this need, increased knowledge and process understanding of the atmosphere are possible.

## 8 Appendix A: SASHe Description

The shortwave array spectroradiometer - hemispheric (SASHe) instrument used in this comparison is one of several

shortwave array spectrometers that were designed and built for ARM through funding associated with the American Recovery and Reinvestment Act (ARRA). The SGP SASHe was installed in March 2011 and has been on site ever since. The SASHe provides measurements of solar irradiance components over the continuous spectral range from UV to the shortwave NIR. It uses a rotating shadowband technique similar to the MFRSR (Sect. 2.2.2) to alternately expose and shade a hemispheric diffuser (shown in Fig. A1a) to direct sunlight, thereby permitting measurements of direct, diffuse, and total

irradiance components over a period of about 30 seconds. Light transmitted through the hemispheric diffuser is routed through a shutter assembly and connected to a pair of commercial Avantes fiber-coupled spectrometers (Fig. A1b) via large-core fused silica fiber. The measurement sequence operated by a laptop PC (Fig. A1c) includes "dark spectra" collected while the shutter is closed followed by spectra collected while the shutter is open and the shadowband is in one of the following positions: (1) below the horizon so the diffuser is exposed to the entire sky, (2) "next to the sun" so that the band

obscures a portion of the sky near the sun and the shadow falls just adjacent to the diffuser, (3) casting a shadow directly across the diffuser, or (4) positioned so that the shadow falls just to the other side of the diffuser.

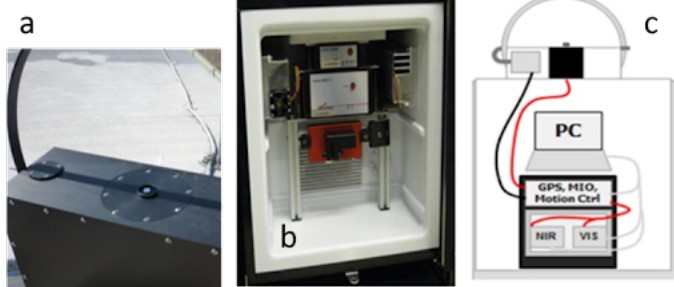

**Figure A1: (a) SASHe optical collector with shadowband casting a shadow over the hemispheric diffuser. (b) The SASHe chiller with spectrometers (top) and shutter (red). (c) Schematic showing collector on outside of building and umbilical connections to the PC, data acquisition, and spectrometers.**

The SASHe calibration includes multiple elements described in more detail in the SASHe Instrument Handbook (Flynn, 2016) summarized here for convenience.

1. Wavelength registration of the spectrometer pixels versus published lines of a Hg-Ar emission lamp.

2. Wavelength resolution varies from ~2.6 nm to ~2.3 nm over the UV/VIS spectral range.

3. Internal straylight levels confirmed to meet vendor specification of <0.1% over most of the spectral range.
Empirical corrections have been applied for the short wavelength region.

4. External straylight leaking through fibre optic jacketing is confirmed to be negligible.

5. Diffuser angular response, aka "cosine correction" has been measured by rotating the SASHe diffuser through the full range of incident angle +/- 90 degrees relative to a broadband light source. This correction is most significant for the direct beam measurement which also incorporates an implicit correction for spectrometer signal non-linearity (discussed below). Effects of the diffuse angular response on the diffuse hemispheric component are modeled based on the measured direct beam correction. Both corrections are applied in routine processing

6. Spectrometer signal linearity. A small but non-negligible non-linearity has been identified. Normal processing of the direct beam signal removes this, but does not remove effects on the diffuse hemispheric component which may contribute to differences observed in this study.

7. Nominal spectral response has been determined by reference to a spectrally calibrated QTH lamp positioned much closer than at the reference distance due to signal strength. This wavelength response curve is used only for pixels at wavelengths that are not amenable to Langley calibration as noted below.

8. Langley calibration is conducted individually for all pixel wavelengths deemed not to be affected by water vapor or strong molecular absorbers as indicated in the SASHe ARM data files. The y-intercept of a Langley plot represents what the instrument would measure at the top of the atmosphere. Dividing the measurement by this amount yields the unitless atmospheric transmittance for each wavelength. Multiplying the unitless transmittance by extraterrestrial or "top of atmosphere" irradiance yields calibrated irradiance components in the same radiometric units as the reference source.

The SASHe data processing is conducted in a few distinct stages. First, conversion of the raw ASCII files generated by the instrument into daily netcdf files. Second, identification of the dark spectra and the irradiance components by analysis of the raw spectra collected through the shadowband sequence, followed by cosine correction and application of a nominal irradiance calibration based on lamp measurements. Third, application of Langley regressions to the log of the direct beam signal from each pixel versus the airmass. Depending on conditions, a maximum of two Langley regressions are possible per
755    day (one before noon and one afternoon) but typical atmospheric variations make these initial calibrations very noisy. Fourth, filtering of several weeks of the initial noisy data with an interquartile filter followed by a sliding Gaussian filter to

obtain daily calibrations that vary by less than a percent on average. Fifth and finally, computation of total optical depth from:

$$\tau = \frac{\ln(\frac{I}{I_o})}{m},$$ (A1)

where $\tau$ is the total optical depth, I is the irradiance direct normal measurement, $I_o$ is the smoothed Langley calibration at the top of the atmosphere, and $m$ is the optical airmass. Aerosol optical depth (AOD) is computed from this by subtracting Rayleigh molecular optical depth (OD) at each wavelength and by subtracting ozone optical depth at affected wavelengths using the column abundance of ozone from OMI (Sect. 2.2.6). ARM processing does not attempt other gas OD corrections but suspect wavelengths are flagged with quality checks. Within the same processing stage, normalized transmittances are computed for each component I (that is, direct normal, direct horizontal, diffuse hemispheric, and total hemisphere) divided by of the top-of-atmosphere calibration Io at the same wavelength. Lastly, each normalized transmittance component is multiplied by the extraterrestrial solar irradiance and adjusted for the earth-sun distance to yield units of W m$^{-2}$ nm$^{-1}$. Cloud screened AODs are obtained using Alexandrov's normalized atmospheric variability method, available in the data files and applied at quality-check flags.

As mentioned in Sect. 2.2.3, instrument issues affected operation of the SASHe during the HSR1 test period that limited the SASHe comparison to clear-sky conditions. The instrument issues included a mechanical issue that led to frequent failure to clearly distinguish the direct solar and diffuse hemispheric irradiance components, which was especially the case for cloudy skies. Additionally, a detector nonlinearity has been identified (but not yet corrected) that affects the diffuse irradiance values and thus also the total irradiance reported by the SASHe.

# 9 Appendix B: PAR Comparison

Photosynthetically active radiation (PAR), integrated $F_{total}$ from 400 to 700 nm, is measured by the PAR Quantum Sensor (PQS1) instrument as part of the Carbon Dioxide Flux Measurement System (CO2FLX) (Chan and Biraud, 2022). The measured PQS1 PAR is compared to the HSR1 PAR in Fig. B1. The mean (relative) difference for the HSR1 PAR compared to the PQS1 PAR is 53.9 (4.7%) μmol m$^{-2}$ s$^{-1}$. Better agreement is found for overcast conditions (-1.0%) and worse agreement is found for clear-sky conditions (6.0%). However, the spread in the PAR comparison is smallest for clear-sky, noting that the correlation coefficient is highest (1.00) and the standard deviation of the difference is smallest (59.4 μmol m$^{-2}$ s$^{-1}$). The spread in the PAR comparison is largely due to partial cloudy-skies and overcast skies as the standard deviation of the differences are larger (~130-210 μmol m$^{-2}$ s$^{-1}$). The larger spread for the cloudy-sky PAR comparison may be partially due to clouds rapidly varying over time and space.

The HSR1 PAR is found by first converting the HSR1 $F_{total}$ from W m$^{-2}$ to μmol m$^{-2}$ s$^{-1}$ to match the PQS1 units by considering a spectral conversion factor ($f$) based on Planck's formula, such that:

$$f = \frac{\lambda}{hcN_a} \; x \; 10^{-3} = 0.00835935 \, \lambda ,$$ (B1)

where $\lambda$ is the wavelength in nm, $h$ is Planck's constant, $c$ is the speed of light, and $N_a$ is Avogadro's number. The spectral HSR1 values in μmol m$^{-2}$ s$^{-1}$ are then integrated from 400 to 700 nm to obtain the HSR1 PAR.

For PQS1 PAR values below ~1000 μmol m$^{-2}$ s$^{-1}$, the collocated PAR observations with the highest frequency align along the 1:1 line with a mean difference of 45.7 μmol m$^{-2}$ s$^{-1}$. Above ~1000 μmol m$^{-2}$ s$^{-1}$, HSR1 PAR values are biased high with a deviation from the 1:1 line and a mean difference of 67.6 μmol m$^{-2}$ s$^{-1}$. However, the largest disagreement values switch from biased high to biased low near values of about 1500 μmol m$^{-2}$ s$^{-1}$. This can be seen in that the 1$^{st}$ (99$^{th}$) percentiles of the differences are -219.8 (807.7) μmol m$^{-2}$ s$^{-1}$ and -917.9 (326.1) μmol m$^{-2}$ s$^{-1}$ for values below and above 795 1500 μmol m$^{-2}$ s$^{-1}$, respectively.

    The PAR comparison was separated into clear-sky, partial cloudy-sky, or overcast times (not shown). The mean (relative) difference is 86.2 (6.0%), 64.3 (4.8%), and -4.9 (-1.0%) μmol m$^{-2}$ s$^{-1}$ for clear-sky, partial cloudy-sky, and overcast, respectively. This aligns with the results presented in Fig. B1 such that better agreement is found at lower values than at higher values, where higher values correspond more towards clear-sky and lower values correspond more so to 800 overcast conditions. While the mean difference is largest for clear-skies, the spread in the comparison is smallest noting that the correlation coefficient is highest (1.00) and the standard deviation of the difference (59.4 μmol m$^{-2}$ s$^{-1}$) is the smallest of the three conditions. This may suggest that for clear-skies the conversion factor is too large or that the HSR1 $F_{total}$ is consistently too high in this spectral range. The spread in the PAR comparison in Fig. B1 is largely due to partial cloudy-skies and overcast skies as the standard deviations of the differences are 210.3 and 129.6 μmol m$^{-2}$ s$^{-1}$, respectively. The 805 larger standard deviations may be partially a consequence of clouds rapidly varying over time and space.

    While the PQS1 is utilized as a reference PAR measurement to evaluate the HSR1 PAR, there is no reported uncertainty for the PQS1 PAR and no traceable accurate reference for PAR measurements. Across different PAR instruments, the reported estimated PAR uncertainty is typically within 5% for ideal conditions but intercomparisons can be up to 20% different even for the same instrument (Mõttus et al., 2012). This suggests that the HSR1 PAR estimates are 810 generally within measurement uncertainties of existing PAR instruments even under different conditions.

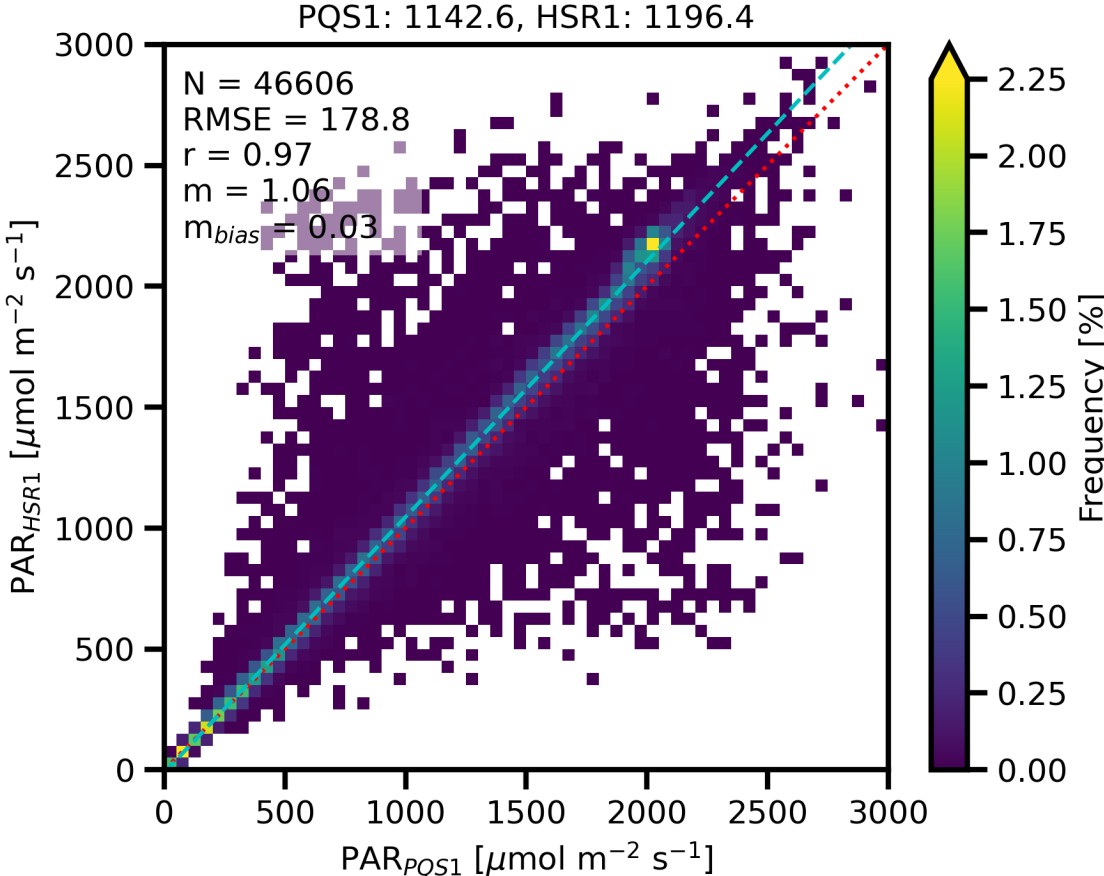

**Figure B1: Frequency histogram for collocated PAR (μmol m⁻² s⁻¹) between the PQS1 and the HSR1. The mean values are given above the plot and the sample size (N), root mean square error (RMSE), correlation coefficient (r), regression line slope (m), and bias regression line slope (m_bias) are shown in the top left. The 1:1 line is indicated by the dotted red line and the regression line is indicated by the dashed light blue line.**

**Data availability**

Data can be downloaded from the ARM data archive (https://www.arm.gov/data/) for the HSR1 (sgphsr1C1.00; http://dx.doi.org/10.5439/1888171), CSPHOT (csphotaodfiltqav3; http://dx.doi.org/10.5439/1461660), MFRSR (sgpmfrsr7nchaod1michC1.c1 and sgpmfrsr7nchaod1michE13.c1; http://dx.doi.org/10.5439/1756632), ozone (gecomiX1.a1; http://dx.doi.org/10.5439/1874262), PAR (sgpco2flxrad4mC1.b1; http://dx.doi.org/10.5439/1313017), and RADFLUX (sgpradflux1longE13.c1; http://dx.doi.org/10.5439/1395157).

## Author contribution

KAB, LDR, JW, CF, AT, and MR conceptualized the study. JW provided the HSR1 instrument and data. JW, AT, and MR provided project administration that facilitated the instrument deployment. CF, LM, GBH, and CH facilitated the operation and data processing of the comparison instruments. KAB and JW performed the formal analysis. KAB, JW, and CF prepared the figures. KAB drafted the manuscript. KAB, LDR, JW, CF, AT, LM, GBH, and CH reviewed and edited the manuscript.

## Competing interests

The authors declare that they have no conflict of interest.

## Acknowledgements

K. A. Balmes and L. D. Riihimaki thank E. Hall and L. Soldo for their help installing the HSR1 in Boulder. The authors also thank the SGP site technicians and all others who helped with the HSR1 test operations and corresponding logistics. This work was funded by the U.S. DOE ARM.

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
