# Peer review of "Evaluation of the hyperspectral radiometer (HSR1) at the ARM SGP site"

_Atmospheric Measurement Techniques, 2023_

## Referee Comment (RC2)

This paper reports on the performance a new radiometer, HSR1, that measures total and diffuse spectral irradiance. HSR1 measured irradiances, diffuse ratio, and derived aerosol optical depths are compared to other spectral and broadband radiometer systems: MFRSRs, SASHe, a Cimel sunphotometer, the RadFlux data product. The instruments operated at the ARM Southern Great Plains site for two months in spring/summer of 2022. The conclusion is that HSR1 measurements of total and diffuse spectral irradiance well in relation to the other instruments, however significant biases exist for irradiance measurements near the tails of the instruments spectral range.

This paper does a reasonable job with the statistical comparison between the HSR1 data and the data collected by the reference instruments at the SGP site. However, for a paper that aims to detail the functionality of a new instrument, key components of the manuscript are lacking. These include: details about the HSR1 instrument hardware, calibration procedures, and the study design that are necessary for an instrument paper of this sort. A discussion of the measurement biases in the context of the unique instrument design features. The role measurement uncertainty has on the analysis is not sufficiently addressed. Further, the quality of the writing, both with respect to grammar and structure of the paper, is not at an appropriate level for publication. I've highlighted a few examples of writing quality issues in the following comments. Given these issues I do not recommend this manuscript for publication in AMT at this time.

Comments:

Lines 23-25 – "The HSR1 quantities are also compared at other wavelengths to the collocated instruments, where similar agreement is found for the spectral irradiances, although relatively larger disagreement is found at higher wavelengths, especially for spectral AODs." To me this sentence reads awkwardly. I recommend that it be reworked.

Line 44 – It is worth pointing out somewhere in the manuscript that HSR1 measures total and diffuse irradiance simultaneously, which is in contrast to rotating shadowband systems.

Line 45-59 – Much of the contents of this paragraph that detail characteristics of HSR1 should be moved out of the introduction to a section that overviews the hardware and calibration procedures of the instrument.

Line 55 – "As the sun moves across the sky throughout the day…" It is also worth mentioning here that the same holds true if the instrument position moves. Again, this seems to be a unique feature of the shadowmask design of the HSR1.

Line 57-59 – "The measured diffuse assumes that the diffuse light is scattered equally angular, i.e., isotropic. The isotropic assumption may not be applicable due to the scattering properties of aerosols and clouds which may have a preferential scattering angle." Here is an example of where the writing quality needs to be improved. This paragraph would read more clearly if the writing were more concise, e.g.: The measurement of diffuse irradiance assumes the scattered

light is isotropic. Then the following discussion of the implications of assuming isotropic diffuse light is important (and necessary), but it does not fit in this section of the paper.

Line 71 – This section describing HSR1 is insufficient. Here is where some of the content from the introduction should go – describe the shadow mask, the specifics of the HSR1 design including a description of the spectrometers used, the theory behind the diffuse and total irradiance measurements, what is new about HSR1 specifically compared to past iterations of the instrument, briefly overview the calculation of the direct irradiance, etc.. Second, how is HSR1 calibrated? At the very end of the manuscript in the Discussion section it is noted that the HSR1 is calibrated against a lamp standard -- why is that procedure not described here? Why is there no discussion of the cosine response of the HSR1? Field-of-view issues, and lensing effects of the dome are briefly mentioned but that discussions lacks detail. It is useful to understand the limitations of the HSR1 instrument so the results of this intercomparison can be interpreted.

Line 74 – what is the native sampling rate of HSR1? Or how many data points are getting smoothed over in 1 minute?

Line 78 – roughly how far apart are the different instruments from each other? It is not easy to infer this from the reported coordinates. A map detailing the locations of the various instruments would be useful.

Line 79 – I would move Figure 1 that shows an example of HSR1 irradiance data to a later part of the paper. Also, why not include the comparable irradiance measurements from the other instruments?

Line 90-95 – This discussion of the cause of the downtime seems unnecessary.

Line 98 – How was it determined that stray light is causing the noise at the tails of the spectrum?

Line 102-103 – the comment on future designs of the HSR1 is better suited for a discussion section later in the paper.

Line 110-111 – "We consider how the dome lensing effect corrected total and diffuse spectral irradiances may affect the results in Sect. 5." I do not believe that this was ever done in Section 5.

Line 115 – throughout the manuscript I am not sure it is necessary to refer to measured or derived quantities at a specific wavelength as "spectral". As far as my knowledge goes this is not standard practice. It is more readable to just say the AOD at 500 nm, for example.

Line 130 – again, I am not sure the discussion of data reprocessing is necessary.

General comment – there needs to be more discussion of the magnitude and sources of measurement uncertainty for HSR1 and the reference instruments. This will help the reader better understand the significance of the differences between the measurements.

Line 145 – how do visible and sub-visible cirrus impact the determination of clear-sky periods? Cirrus can significantly bias the diffuse irradiance measurement.

Line 149 – what does this manuscript gain by including the comparison of PAR? I recommend removing this portion of the analysis.

Line 170 – it should be made clear that in equation 2 the optical depths have a spectral dependence.

Line 192 – what is the rational for kicking out half of the data points when deriving the TOA DNI?

Line 227 – "Therefore, portions of the surface downwelling diffuse light are not measured by the HSR1 and…" It seems like this light is measured by the HSR1, but it is just attributed to being direct irradiance?

Line 228 – Throughout the paper comparisons are done between the various reference instruments. As currently written, this seems unnecessary as I do not see what value it adds to the analysis, and it distracts from the main topic which is the evaluation of the HRS1.

Line 251 – In Figure 4 why not also include the direct irradiance?

Line 255 – This table is hard to read and interpret and does not hold a lot of utility to the reader. Much of this information is already stated in the text, so I'd either omit the table or present the data in a graphical format.

Line 271-273 – "Similar to the total spectral irradiance, the MFRSR C1 and MFRSR E13 diffuse spectral irradiance comparison at 940 nm is the largest relative difference, which is nearly an order of magnitude larger than all other wavelengths (0.9-1.9%). This further highlights the challenges in measuring the spectral irradiance at 940 nm." I found this wording confusing and suggest it be revised.

Line 279 – the details about the MFRSR spectral channel widths seems better suited for Section 2.

Line 287 – what is the motivation for comparing HSR1 to SASHe under clear sky-condtions. As is this manuscript is only presenting statistical quantities of HSR1 versus other instruments with little justification for doing so or interpretation of the results. For example, how might the shadowmask design of HSR1 influence this comparison with a shadowband type instrument?

Line 304-312 – again, what about the instruments or experimental setup is driving these differences.

Line 325 – this sentence should be reworded: AODs are not collected, they are calculated.

Line 340 – again, I do not see the value in the comparison of the AOD derived from the reference instruments.

Line 375 – it may be worthwhile to include a timeseries figure or two of irradiance and AOD that illustrates under what solar zenith angle and cloud conditions there is good and poor agreement between HSR1 and the reference instruments.

Line 423 – "Noting the measurement uncertainty of ±3% in the diffuse flux (Michalsky and Long, 2016), only 16.5% (all times) and 18.3% (clear-sky times) of the diffuse flux errors due to considering the HSR1 diffuse ratio are within measurement uncertainty." I had a hard time understanding this sentence and would recommend rewording it.

Line 509 – This section is not a discussion section but it is a summary. Here is a good place to discuss how the design of the HSR1 impacts its ability to measure irradiance relative to the reference instruments. Under what conditions does is perform well (e.g., clear-sky, cloudy-sky)? And when there are biases in the data HSR1 produces, why? For example, what impact does the wide field-of-view, the cosine response of the sensor, the assumption that the diffuse light is isotropic, etc., have on the measurements.

---

## Author Response (AR1)

Dear Dr. Schmidt,

Thank you for guiding the review of our manuscript entitled " Evaluation of the hyperspectral radiometer (HSR1) at the ARM SGP site" (doi: 10.5194/amt-2023-115).

The comments and suggestions from the reviewers were constructive and valuable to improving the manuscript. We are now submitting a revised manuscript in which we have addressed all the reviewer's comments and suggestions. The point-by-point responses are included below with the reviewer's comments in black and our replies in blue. The page and line numbers correspond to the change accepted version of the manuscript (i.e., "clean").

Thank you for your consideration of this manuscript.

Sincerely,
Dr. Kelly Balmes
* * *
Reviewer #1
Referee comment on "Evaluation of the hyperspectral radiometer (HSR1) at the ARM SGP site" by Balmes et al.

The study summarizes the results of a comparison of spectroradiometer measurements over a two-month period. Data from an instrument capable of measuring total and diffuse spectral irradiance separately are compared with several other instruments in terms of total and diffuse irradiance, diffuse fraction, and derived AOD.

General Comments

The authors provide a comprehensive overview of statistical measures used to describe the correlations and regressions between the different data sets. Such intercomparisons as part of an instrument study are important to document the suitability of new instruments. However, this work is limited to the mere presentation of statistical comparative figures without deeper discussion and investigation of the causes. For this reason, I cannot recommend the manuscript in its current form for publication in ATM, but would encourage the authors to submit a new manuscript that is less descriptive. Below I give comments and some suggestions to improve and expand the content.

Thanks for the reviewer's efforts to review this paper. See our replies below.

Specific Comments
1. It is not clear what the new technical specifications of the HSR1 are compared to the instrumentation described in Wood et al. (2017) and Nogren et al. (2022)? Is it the same instrument? The instrumental design of the HSR1 is not fully described by the authors. What kind of spectrometers are used?

We have revised the text to include more specific details on the technical specifications of this instrument and what is different from previous prototypes and references (P. 3, L71-76): "The spectrometer within the HSR1 is a significant improvement over those reported in Wood et al. (2017), which used either an array of low-cost commercial spectrometers, or a fibre switch with a higher specification spectrometer to measure the seven spectral inputs. The current HSR1 uses a custom designed multichannel spectrometer, which images and spectrally disperses the light from the input fibres onto a 2D image sensor, so all channels are measured simultaneously. This significantly improves the measurement resolution, speed, and matching between the channels compared with the earlier implementations. An early version of this system was also used by Norgren et al. (2022)."

2. The study lacks a detailed uncertainty analysis. The authors provide little information on calibration issues and only mention the "dome lensing effect" in the discussion section without explaining it. They should definitely expand this section. What are the single measurement / calibration uncertainties and how do they contribute to the different products?

We have revised the text to include information on the HSR1 uncertainty (P8, L137-140): "A reference HSR1 is calibrated by removing the shading mask, and exposing the sensors to a 1000 W 'FEL' lamp, with an output spectrum calibrated by the UK NPL. This calibration is transferred to other HSR1s during routine calibrations and calibration checks using an integrating sphere. The expected uncertainty in $F_{total}$ measurements is expected to be around 5% between 400 nm and 900 nm."

We have also added in discussion on further sources of measurement/calibration uncertainties in Section 5. We also plan to discuss further in a follow-on paper of post-processing modifications and other sources of measurement/calibration uncertainties.

3. The presentation of the different instruments in Section 2 should also include the instrumental uncertainty. The subsubsections (Sec. 2.2) that inform about the other instruments are quite short. Consider summarizing them without subdividing them further and showing a table with the main specifications.

Thank you for the suggestion. We have added in Table 1 (see below) which summarizes the main instrument specifications and uncertainties to Section 2.

Table 1. Instrument specifications including spectral range, spectral resolution, retrieved quantities, and uncertainty estimates.

| Instrument | Measurement | Spectral coverage (resolution) | Retrieved quantities | Uncertainty estimates |
|---|---|---|---|---|
| HSR1 | Total and diffuse hyperspectral irradiances | 360-1100 nm (3 nm) | AOD at 415, 440, 500, 615, 673, 675, and 870 nm | Total irradiances: 5% AOD: 0.02 |
| CSPHOT | Direct solar irradiance and sky radiance | 340, 380, 440, 500, 675, 870, 1020, and 1640 nm | AOD at 440, 500, 675, and 870 nm | AOD: 0.01 |

| MFRSR | Total and diffuse spectral narrowband irradiances | 415, 500, 615, 673, 870, and 940 nm | AOD at 415, 500, 615, 673, and 870 nm | Irradiances: 3% AOD: 0.01 |
|-------|---------------------------------------------------|-------------------------------------|---------------------------------------|---------------------------|
| SASHe | Total and diffuse hyperspectral irradiances | 336 to 1100 nm (~2.5 nm), 950 to 1700 nm (6 nm) | AOD at 415, 500, 615, 673, and 870 nm | AOD: 0.02-0.03 Irradiances: AOD relative uncertainty multiplied by the airmass |

4. P2L46: "radiometer with no moving parts is now available called the hyperspectral radiometer (HSR1)" – I would leave out the phrase "no moving parts". It would be better to say that no rotating shadow band is required.

We have revised the text to make clear that no rotating shadow band is required (P. 2, L45-47): "In an attempt to ease the operational difficulties of hyperspectral radiometry, a newly developed hyperspectral radiometer with no moving parts and no requirement for rotating shade rings or motorized solar tracking devices is now available called the hyperspectral radiometer (HSR1) (Wood et al., 2017; Norgren et al., 2022)."
We have kept in the text that no moving part since this is a unique feature for measuring diffuse that provides utility for remote and harsh deployments.

5. P2L53-L59: I had difficulty understanding the brief description of how to separate total and diffuse irradiance. Only the references given made it clearer. Even though it is redundant to the other publications, I would recommend a sketch for better understanding. Since these are instrumental details, I would place this information in Sec. 2.

We have revised the text to provide more details on how the total and diffuse spectral irradiances are measured based on the multiple sensors with equations in the introduction (P. 2-3, L53-70): "The HSR1 was designed with seven spectral sensors: six sensors placed on a hexagonal grid, one sensor at the center, under a complex static shading mask (see Figs. 1 in Badosa et al., 2014 and Wood et al., 2017). The shading mask design is to ensure that, at any time, for any location: (1) at least one sensor is always exposed to the full solar beam; (2) at least one sensor is always completely shaded and; (3) the solid angle of the shading mask is equal to $\pi$ thus corresponding to half of the hemispherical solid angle. With no moving parts or specific azimuthal alignment, the instrument is ideal for deployment on moving platforms such as ships and remote locations where regular maintenance is difficult.
    Assuming isotropic diffuse sky radiance, the third property related to the shading mask implies that all sensors receive equal amounts (50%) of $F_{diffuse}$ from the rest of the sky hemisphere. Therefore, at any instant, the minimum signal ($F_{min}$) measured among the seven sensors is the shaded sensor, which measures half the $F_{diffuse}$, and the maximum signal ($F_{max}$) from among the seven sensors is fully exposed to the solar beam, and therefore measures the direct irradiance ($F_{direct}$) plus half the $F_{diffuse}$. From this the following relationships can be formed:
$F_{diffuse} = 2F_{min},$ (1)

$F_{direct} = (F_{max} - F_{min})$,                                                                  (2)

$F\_total = F_{direct} + F_{diffuse} = F_{max} + F_{min}$.                                     (3)

In the HSR1, $F_{max}$ and $F_{min}$ are selected from the integrated spectral measurements from each sensor, and these relationships are applied to the corresponding spectral measurements to calculate the Ftotal and $F_{diffuse}$. Due to the nature of the measurements, the $F_{total}$ and $F_{diffuse}$ are measured simultaneously. This is in contrast to rotating shadowband systems which must make the $F_{total}$ and $F_{diffuse}$ measurements separately and, therefore, at different times."

We have kept this information in the introduction following a suggestion from the editor's initial review.

6. P2L53: "the shadow pattern allows one of the seven sensors to be illuminated unobstructed by the shadow pattern, which measures the total irradiance" – According to Wood et al. (2017) it should be I_max+I_min, which gives the total irradiance. Please clarify.

We have revised the text to clarify how the total irradiance is calculated, including the equations (Section 1). See above reply.

7. P3L79-L83 + Fig. 1: I think that this example plot is not needed at this point. Data coverage is reported at the beginning of this section. All the detailed information about the reasons for the downtime may be less important to the reader. Try to shorten them. Perhaps show a time series of the radiation data along with the cloud cover data.

We have shortened the HSR1 downtime information by removing the paragraph description and only including in the text (P. 3, 90-91): "The HSR1 exhibited excellent uptime and near-autonomous data collection over the two-month test period with an uptime of 97.5%."
We have also added in a new Figures 4 and 5 that is a timeseries of the irradiance(500 nm) and AOD(500 nm) with a clear-sky marker.

8. The second part of Sec. 2.1 should contain more technical details of the HSR1 instrument. Here the spectra from Fig. 1 would fit.

We have added in more technical details to the second part of Section 2.1. We have removed the downtime information description and, therefore, the spectra figure has not moved but is now where the text of technical details is.

9. P5L98-L103: The spectral range limitation could be better justified by using radiative transfer simulations that show the low performance at the edges of the spectral range. Is it really stray light that is causing the low performance? Have you done lab tests with edge filters to analyze this?

We have revised the text to include that the straylight issues are known based on lab tests (P. 7, L118-120): "In particular, considerable noise was noted for wavelengths greater than 1000 nm (Fig. 2c) as the measurements were contaminated by second-order stray light as identified in the lab using a monochromator. As with all spectrometers, measurements at the two extremes of the spectrum have low sensitivity, and therefore additional noise is apparent."

10. P5L108: "lensing effect" might be important – Please elaborate. It is referred to Sec. 5, but there is no deeper discussion.

We have added in further discussion of the dome lensing effect in the discussion section (P. 36, L672-675): "The effects of correcting for the dome lensing variability first noted in Badosa et al. (2014) will also be investigated further, and may reduce some of the variabilities in the $F_{total}$ and $F_{diffuse}$, and retrieved AOD. Initial analysis indicates that the dome lensing effect on the results in this study are small with a change of 0.01 or less in the $F_{total}$, $F_{diffuse}$, and AOD at 500 nm." We also plan to discuss further in a follow-on paper of post-processing modifications and other sources of measurement/calibration uncertainties.

11. Section 2.2: Please give uncertainties for all instruments / products.

Thank you for the suggestion. We have added uncertainty estimates into the text where available as well as summarized in new Table 1.

12. P6L143: What is the wavelength range that is covered by the instrument?

We have added into the text the spectral range that is covered by the SW instruments that are input into RADFLUX (P. 12, L183-184): "The SW broadband radiometer spectral range is 295 to 3000 nm (Andreas et al., 2018)."

Andreas, A., Dooraghi, M., Habte, A., Kutchenreiter, M., Reda, I., and Sangupta, M.: Solar Infrared Radiation Station (SIRS), Sky Radiation (SKYRAD), Ground Radiation (GNDRAD), and Broadband Radiometer Station (BRS) Instrument Handbook. Ed. by Robert Stafford, ARM Climate Research Facility. DOE/SC-ARM-TR-025. https://doi.org/10.2172/1432706, 2018.

13. P6 Sec.2.2.3: Too many details on instrumental failures. It is sufficient to say, that only cloudless conditions could be considered due to instrumental issues.

Thank you for the suggestion. We have revised the text to reduce the detail of the SASHe data reprocessing (Section 2.2.3).

14. P7L150: I am not sure that the comparison of PAR data is really necessary in this study, since it is just another quantity based on spectral integration.

The PAR comparison shows an application of the HSR1 that is possible with hyperspectral information. We have moved the PAR comparison to new section Appendix B.

15. 3: Please discuss the uncertainty of the AOD retrieval.

Thank you for the suggestion. We have added in discussion of the HSR1 AOD retrieval (P. 11-12, L236-241): "The HSR1 AOD uncertainty is quantified. Since the HSR1 AOD is retrieved from Langley regressions, the AOD uncertainty is independent of the HSR1 irradiance calibration. The HSR1 AOD uncertainties are due to: (1) uncertainties in the TOA DNI, (2) cosine errors, and (3) dome lensing effects. The TOA DNI uncertainty is 1% as determined by the standard error of the means. The cosine error uncertainty is 2% based on instrument design limits. The dome lensing effect uncertainty is 1% as calculated from optical theory. The HSR1 AOD uncertainty is determined by considering the perturbation of the HSR1 AOD to the uncertainty sources. The resultant perturbation to the HSR1 AOD is ±0.02."

16. P8L183: How does tau_gas depend on the vertical profile of temperature and pressure? Is ozone the only type of gas that contributes?

The AOD retrieval in this study only considers the contribution of ozone columnar amount. We have updated the text to make this point clear (P. 11, L222-224): "For $\tau_{gas}$, only the effect of ozone is considered due to the wavelengths considered as other gaseous absorption is considered negligible (Koontz et al., 2013; Ermold et al., 2013). In addition, only the column amount of ozone is considered (i.e., no vertical dependence)."

17. The figures are well presented but include all formula signs in the text. Example Fig. 2: "F_total".

Thank you for the suggestion. We have revised the text to change to formula signs throughout.

18. All frequency histograms give the mean value of the two-month period. I am not sure if this is an appropriate measure since the data are not normally distributed and may have multiple modes.

The frequency histograms along with the regression lines, especially the regression line of the bias in the subplots, provide an evaluation across the distribution beyond the means. We have also added in the root mean square error to all plots for another evaluation metric. See Figures 5-12 and B1. Furthermore, the regression line of the bias helps capture how the bias changes across multiple modes (P. 15, L263-266): "In addition, the regression lines and the regression lines of the bias are shown in Figs. 6 and 7, which provides additional information on how the bias changes across different modes. The regression lines of the bias are constructed by regressing the bias (e.g., instrument 2 - instrument 1) with the reference instrument values (e.g., instrument 1)."

19. 2d: Only the regression line is shown here. What does the scatter plot look like?

The frequency plot of the bias is shown below. The regression lines are shown to summarize the bias tendencies. The frequency plot of the bias is not shown in the manuscript as similar information is shown in Figure 6a-c.

[Figure]

20. All scatter plots / frequency histogram: Is the scatter between the different instruments within the measurement uncertainties?

We have added in text on the comparison of the instruments in terms of the uncertainties throughout Sect. 4.

21. 4: same as Comment#16. To show a possible wavelength dependence, it might be helpful to plot the RMSE as a function of wavelength. Table 1 is sufficient for the interpretation of the sign of the bias.

Thank you for the suggestion. We have added in a plot of the RMSE as a function of wavelength to Figs. 8d (previously Fig. 4). See revised figure below.

[Figure]

22. Table 1: Way too many numbers. I recommend reducing the data to the HSR1 comparison only. The AOD results should be given in a separate table in Sec. 4.2.

We have removed previous Table 1 to improve readability.

23. 4.1.1: I don't see any gain in information when the comparison results between the other instruments are shown. It is a bit monotonous to give all the numbers in the text which can be read from the table. The same holds for the AOD comparison P18L340-L351.

The comparison results between the other instruments provide a reference for the HSR1 comparison. We have revised the text to highlight this point for why the other instruments are compared. For the irradiance, the text is revised to (P. 15, L261-263): "The MFRSR C1 and

MFRSR E13 spectral irradiances are also compared to each other in Figs. 6-8 to provide context to the HSR1 comparison by showing the level of agreement between two instruments of the same model at the same location." For the AOD comparison, the text is revised to (P. 24, L429-430): "This comparison provides context to the HSR1 AOD comparison by quantifying the level of agreement between established instruments and AOD retrievals." We have also removed the tables and reduced the text to clarify the story.

24. P21L363: "The spectral AOD results at all wavelengths are similar to those at 500 nm (Fig. 6)" – rewrite. The absolute numbers are not similar.

We have revised the text to make clear that the relative ordering of the comparison is the same across the wavelengths and not the numbers (P. 26, L461-463): "The relative ordering in the AOD comparison at all wavelengths are similar to those at 500 nm (Fig. 10): the mean HSR1 AOD is larger than those from the CSPHOT and the two MFRSRs except for the mean SASHe AOD, which is larger than the mean HSR1 AOD."

25. 7: same as Comment 21: To show a possible wavelength dependence, it might be helpful to plot the RMSE as a function of wavelength.

Thank you for the suggestion. We have added in a plot of the RMSE as a function of wavelength to Figs. 11i-j (previously Fig. 7).

[Figure]

26. Maybe swap Sec. 4.3.1 and Sec. 4.3.2. It would make more sense to look at the diffuse spectral components first before showing the integrated values.

Thank you for the suggestion. We have swapped Section 4.3.1 and Section 4.3.2 in order to present the diffuse spectral diffuse ratio comparison before the broadband diffuse ratio comparison.

27. P22L408: "The motivation of this comparison is to understand if the HSR1 integrated diffuse ratio captures the diffuse ratio in the absence of a diffuse broadband irradiance observation (e.g., only total broadband SW measurements) despite measuring only a portion of the solar spectral range." The wording could be improved. Do you mean "broadband" in the sense of solar broadband? The spectral integration of the measured total and diffuse irradiance gives a broadband irradiance but not fully covers the solar spectral range. To identify the missing fraction could be analyzed more deeply by using a radiative transfer model.

We have revised the text to clarify that it is the solar broadband (Section 4.3.2). We agree that a radiative transfer model could identify the missing fraction. However, our intent is to assess how well the diffuse ratio measured by the HSR1 compares to those that are measured by solar broadband instruments.

28. P23L412: A lower mean diffuse ratio is reported for the HSR1 than derived from the Radflux instrument which covers a broader spectral range. I would expect it to be the other way around, since with increasing wavelength the diffuse ratio decreases strongly, as radiative modeling could show.

Yes, the low bias in the HSR1 diffuse ratio is due to the low bias in the HSR1 diffuse measurement, which is opposite of the high bias induced by the smaller spectral range of the HSR1 compared to Radflux. We have added into the text clarification that the smaller spectral range would introduce the opposite bias observed (P. 31, L576-578): "Furthermore, the smaller solar spectral range of the HSR1 would induce a high bias as the diffuse ratio decreases with increasing wavelength. This further suggests that the low bias in the HSR1 diffuse measurements is the dominant feature for the low diffuse ratio bias."

29. P23L422: "The mean diffuse flux error is -16.7 and -7.9 W m-2 for all times and clear-sky times, respectively." Perhaps it would be better to show a distribution of the bias illustrating the different modes.

Thank you for the suggestion. We have revised the figure from a boxplot to relative frequency plots in new Figure 12 c&d (see below).

[Figure]

30. P23L423: "Noting the measurement uncertainty of ±3% in the diffuse flux (Michalsky and Long, 2016), only 16.5% (all times) and 18.3% (clear-sky times) of the diffuse flux errors due to considering the HSR1 diffuse ratio are within measurement uncertainty." I do not understand this sentence. Please rephrase.

We have revised this sentence for clarity (P. 31, L570-572): "The measurement uncertainty of the $F_{broadband,diffuse}$ is ±3% (Sect. 2.2.4). If the $F_{broadband,diffuse}$ is determined by the $DR_{HSR1}$, then the $F_{diffuse,\,error}$ considering the $DR_{HSR1}$ are within the $F_{broadband,diffuse}$ measurement uncertainty only 16.5% (all times) and 18.3% (clear-sky times) of the time."

31. P25L471: "The SASHe clear-sky spectral diffuse ratios were also compared at 415, 615, 673, and 870 nm. The results are found to be similar to the 500 nm results." – rephrase. The absolute numbers are different.

Thank you for the suggestion. We have revised the text for clarity (P. 29, L535-536): "The relative differences at other wavelengths are found to be similar to the 500 nm relative differences."

32. Discussion section: It is not really a discussion of the results. Definitely more content and deeper thinking about the reasons, uncertainties, and relationships between the quantities is needed here. The information on calibration and post- processing is quite vague.

We have expanded the discussion section to include further discussion. Please see Sect. 5.

33. Summary section: This section is a way to long. It repeats all the numbers which is kind of exhausting for the reader. Try to reduce it to the main points.

Thank you for the suggestion. We have removed the summary section and put the main points in the results sections.

34. Conclusion section: I do not find any conclusion here. What can the reader learn from this study?

We have expanded the conclusion section. Please see Sect. 6 (originally Sect. 7).

Technical Comments
1. P3L78: "C1 (36.607322 °N, 97.487643 °W) and E13 (36.604937 °N, 97.485561 °W)." – Give the distance.

Thank you for the suggestion. We have added a map of the instruments' locations across the site in new Figure 1 (see below), which indicates that the physical distance is 170 m or less.

[Figure]

2. P7L158: "Ozone satellite": rephrase

We have rephrased "Ozone satellite" to "OMI" (Section 2.2.6) to follow the styling of the other data sections.

3. P7L151: "PQS1": What does it mean?

PQS1 is the instrument name, which stands for PAR Quantum Sensor. We have revised the text to include this information (Appendix B).

4. 4: Symbols and labels are too tiny.

We have revised Figure 7 (see below) by increasing the size of the symbols and increasing the font size of the labels.

[Figure]

5.  P15L247: "MFRSR filter" à MFRSR narrowband filter

Thank you for the suggestion. We have revised the text from "MFRSR filter" to "MFRSR narrowband filter" (P. 9, L158).

6.  8c: No y-axis label.

We have added in labels for the x-axis and y-axis for new Figure 13 c&d (previously Figure 8c). See revised figure below.

[Figure]

7. P23L418: Here and at other locations the term "flux" is used. Keep using the term irradiance.

Thank you for catching this. We have replaced "flux" with "irradiance" at this point in the text (Section 4.3.2) as well as throughout.

8. P24L437: "F" irradiance in italic

We have revised the text to italicize the "F" throughout.

Reviewer #2

This paper reports on the performance a new radiometer, HSR1, that measures total and diffuse spectral irradiance. HSR1 measured irradiances, diffuse ratio, and derived aerosol optical depths are compared to other spectral and broadband radiometer systems: MFRSRs, SASHe, a Cimel sunphotometer, the RadFlux data product. The instruments operated at the ARM Southern Great Plains site for two months in spring/summer of 2022. The conclusion is that HSR1 measurements of total and diffuse spectral irradiance well in relation to the other instruments, however significant biases exist for irradiance measurements near the tails of the instruments spectral range.

This paper does a reasonable job with the statistical comparison between the HSR1 data and the data collected by the reference instruments at the SGP site. However, for a paper that aims to detail the functionality of a new instrument, key components of the manuscript are lacking. These include: details about the HSR1 instrument hardware, calibration procedures, and the study design that are necessary for an instrument paper of this sort. A discussion of the measurement biases in the context of the unique instrument design features. The role measurement uncertainty has on the analysis is not sufficiently addressed. Further, the quality of the writing, both with respect to grammar and structure of the paper, is not at an appropriate level for publication. I've highlighted a few examples of writing quality issues in the following comments. Given these issues I do not recommend this manuscript for publication in AMT at this time.

Thanks for the reviewer's efforts to review this paper. See our replies below.

Comments:
Lines 23-25 – "The HSR1 quantities are also compared at other wavelengths to the collocated instruments, where similar agreement is found for the spectral irradiances, although relatively larger disagreement is found at higher wavelengths, especially for spectral AODs." To me this sentence reads awkwardly. I recommend that it be reworked.

We have revised the text to improve clarity (P. 1, L23-25): "The comparisons are within ~10% for the spectral irradiances, except for 940 nm where there is relatively larger disagreement. The AOD comparisons are within ~10% at 415 and 440 nm, however, a relatively larger disagreement in the AOD comparison is found for higher wavelengths."

Line 44 – It is worth pointing out somewhere in the manuscript that HSR1 measures total and diffuse irradiance simultaneously, which is in contrast to rotating shadowband systems.

Thank you for the suggestion. We have revised the text to make this point (P. 3, L68-70): "Due to the nature of the measurements, the $F_{total}$ and $F_{diffuse}$ are measured simultaneously. This is in contrast to rotating shadowband systems which must make the total and diffuse measurements separately and therefore at different times."

Line 45-59 – Much of the contents of this paragraph that detail characteristics of HSR1 should be moved out of the introduction to a section that overviews the hardware and calibration procedures of the instrument.

We have decided to keep this information in the introduction following a suggestion from the editor's initial review.

Line 55 – "As the sun moves across the sky throughout the day..." It is also worth mentioning here that the same holds true if the instrument position moves. Again, this seems to be a unique feature of the shadowmask design of the HSR1.

Thank you for the suggestion. We have revised the text to clarify this point (P. 2, L53-58): "The HSR1 was designed with seven spectral sensors: six sensors placed on a hexagonal grid, one sensor at the center, under a complex static shading mask (see Figs. 1 in Badosa et al., 2014 and Wood et al., 2017). The shading mask design is to ensure that, at any time, for any location: (1) at least one sensor is always exposed to the full solar beam; (2) at least one sensor is always completely shaded and; (3) the solid angle of the shading mask is equal to π thus corresponding to half of the hemispherical solid angle. With no moving parts or specific azimuthal alignment, the instrument is ideal for deployment on moving platforms such as ships and remote locations where regular maintenance is difficult."

Line 57-59 – "The measured diffuse assumes that the diffuse light is scattered equally angular, i.e., isotropic. The isotropic assumption may not be applicable due to the scattering properties of aerosols and clouds which may have a preferential scattering angle." Here is an example of where the writing quality needs to be improved. This paragraph would read more clearly if the writing were more concise, e.g.: The measurement of diffuse irradiance assumes the scattered light is isotropic. Then the following discussion of the implications of assuming isotropic diffuse light is important (and necessary), but it does not fit in this section of the paper.

We have revised the text to clarify this point (Section 1). We have decided to keep this information in the introduction following a suggestion from the editor's initial review.

Line 71 – This section describing HSR1 is insufficient. Here is where some of the content from the introduction should go – describe the shadow mask, the specifics of the HSR1 design including a description of the spectrometers used, the theory behind the diffuse and total irradiance measurements, what is new about HSR1 specifically compared to past iterations of the instrument, briefly overview the calculation of the direct irradiance, etc.. Second, how is HSR1 calibrated? At the very end of the manuscript in the Discussion section it is noted that the HSR1 is calibrated against a lamp standard -- why is that procedure not described here? Why is there no discussion of the cosine response of the HSR1? Field-of-view issues, and lensing effects of the dome are briefly mentioned but that discussions lacks detail. It is useful to understand the limitations of the HSR1 instrument so the results of this intercomparison can be interpreted.

We have revised the text (Section 1 and 2) by adding in more details and specifics on the HSR1 including the shadowmask, spectrometers, theory behind the diffuse and total measurements, what is new to this instrument design, calibration information, cosine response, and field of view.

Line 74 – what is the native sampling rate of HSR1? Or how many data points are getting smoothed over in 1 minute?

We have revised the text to include more details on the sampling rate and temporal resolution (P. 5, 100-105): "The HSR1 spectrometer achieves an optical resolution of 3 nm over the range 350 nm to 1050 nm, and can take a measurement in as little as 200 ms. However, to improve the dynamic range of the instrument over the spectral range, and also capture the range of diurnal irradiances, readings are taken over a series of different integration times, and merged into a single high-dynamic-range measurement. This typically gives a measurement time of around 1 s. There is a trade-off between speed and dynamic range. In this study, measurements were made every 10 s, then averaged and stored every minute to match common solar radiation datasets."

Line 78 – roughly how far apart are the different instruments from each other? It is not easy to infer this from the reported coordinates. A map detailing the locations of the various instruments would be useful.

Thank you for the suggestion. We have added in a map of the instruments' locations across the site in new Figure 1 (see below), which indicates that the physical distance is 170 m or less.

[Figure]

Line 79 – I would move Figure 1 that shows an example of HSR1 irradiance data to a later part of the paper. Also, why not include the comparable irradiance measurements from the other instruments?

Thank you for the suggestion. We have revised Section 2.1 so now Figure 2 (previously Figure 1) is later in the text. We have also updated Figure 2 to include the irradiance measurements from the MFRSRs and SASHe (see revised figure below). Note that the figure has been updated to a different time when all instruments were available.

[Figure]

Line 90-95 – This discussion of the cause of the downtime seems unnecessary.

We have shortened the HSR1 downtime information by removing the paragraph description and only including in the text (P. 3, L90-91): "The HSR1 exhibited excellent uptime and near-autonomous data collection over the two-month test period with an uptime of 97.5%."

Line 98 – How was it determined that stray light is causing the noise at the tails of the spectrum?

We have revised the text to include that the straylight issues are known based on lab tests (P. 7, L118-120): "In particular, considerable noise was noted for wavelengths greater than 1000 nm (Fig. 2c) as the measurements were contaminated by second-order stray light as identified in the lab using a monochromator. As with all spectrometers, measurements at the two extremes of the spectrum have low sensitivity, and therefore additional noise is apparent."

Line 102-103 – the comment on future designs of the HSR1 is better suited for a discussion section later in the paper.

Thank you for the suggestion. We have moved this text to the discussion section (Section 5).

Line 110-111 – "We consider how the dome lensing effect corrected total and diffuse spectral irradiances may affect the results in Sect. 5." I do not believe that this was ever done in Section 5.

We have added in further discussion of the dome lensing effect in the discussion section (P. 36, L672-675): "The effects of correcting for the dome lensing variability first noted in Badosa et al. (2014) will also be investigated further, and may reduce some of the variabilities in the $F_{total}$ and $F_{diffuse}$, and retrieved AOD. Initial analysis indicates that the dome lensing effect on the results in this study are small with a change of 0.01 or less in the $F_{total}$, $F_{diffuse}$, and AOD at 500 nm." We also plan to discuss further in a follow-on paper of post-processing modifications and other sources of measurement/calibration uncertainties.

Line 115 – throughout the manuscript I am not sure it is necessary to refer to measured or derived quantities at a specific wavelength as "spectral". As far as my knowledge goes this is not standard practice. It is more readable to just say the AOD at 500 nm, for example.

Thank you for the suggestion. We have revised the text to remove "spectral AOD" and instead refer to AOD at specific wavelengths. We have kept "spectral irradiance" to distinguish from broadband irradiance.

Line 130 – again, I am not sure the discussion of data reprocessing is necessary.

Thank you for the suggestion. We have revised the text to reduce the detail of the SASHe data reprocessing (Section 2.2.3).

General comment – there needs to be more discussion of the magnitude and sources of measurement uncertainty for HSR1 and the reference instruments. This will help the reader better understand the significance of the differences between the measurements.

We have added into the text uncertainty estimates where available (Section 2). We have also added in discussion of the measurement differences in the context of the uncertainty throughout the results section (Section 4).

Line 145 – how do visible and sub-visible cirrus impact the determination of clear-sky periods? Cirrus can significantly bias the diffuse irradiance measurement.

We have added into the text an expanded description of RADFLUX to clarify how clear-sky periods are identified (P. 10, L186-191): "RADFLUX processing first identifies clear sky time periods using the magnitude and variability of the $F_{broadband, diffuse}$ and $F_{broadband, total}$ that have been normalized to remove the impacts of the diurnal cycle. Clear sky estimates are determined at all times using empirical fits to those data points (Long & Ackerman, 2000). Finally, cloud fraction (CF) is calculated based on a relationship with the normalized diffuse cloud effect (Diffuse measured - diffuse clear sky/total clear sky). Care is taken to distinguish between optically thin and thick clouds in the CF calculations using statistics on the magnitude and variability of the irradiance measurements and the diffuse ratio (see Long et al. 2006 for more details)."

Because the RADFLUX clear sky identification methods are based on the variability of the $F_{broadband, diffuse}$, they capture optically thin cirrus quite well. It compares well to both sky imagers and human observers in its ability to identify optically thin clouds. It is possible that sub-visible cirrus that doesn't have a significant impact on the variability of the diffuse will not be captured in the clear sky estimates. However, if subvisible cirrus is missed in the HSR1 AOD retrieval, then it is also likely missed in all of the AOD retrievals as they all use some kind of measure of the scatter of SW irradiance to determine when skies are cloud-free.

Line 149 – what does this manuscript gain by including the comparison of PAR? I recommend removing this portion of the analysis.

The PAR comparison shows an application of the HSR1 that is possible with hyperspectral information. We have moved the PAR comparison to new section Appendix B.

Line 170 – it should be made clear that in equation 2 the optical depths have a spectral dependence.

We have updated Equations 4 and 5 to indicate that the optical depths have a spectral dependence by adding in that these quantities are a function of wavelength, $\lambda$ (P. 10-11, L208-212):

$$DNI(\lambda) = DNI_0(\lambda)exp\left[-\left(\tau_{Rayleigh}(\lambda) + \tau_{aerosol}(\lambda) + \tau_{gas}(\lambda)\right)m\right] \tag{4}$$

$$ln\left(DNI(\lambda)\right) = ln\left(DNI_0(\lambda)\right) - \left(\tau_{Rayleigh}(\lambda) + \tau_{aerosol}(\lambda) + \tau_{gas}(\lambda)\right)m \tag{5}$$

Line 192 – what is the rational for kicking out half of the data points when deriving the TOA DNI?

We consider the interquartile range to eliminate outliers and reduce noise as is done by other AOD retrievals that the retrieval in this study is based on. We have revised the text to make this point clearer (P. 11, L232-234): "The TOA DNI are then filtered by only considering the interquartile range (i.e., 25th-75th percentile) to eliminate outliers and reduce noise (Koontz et al., 2013; Ermold et al., 2013)."

Line 227 – "Therefore, portions of the surface downwelling diffuse light are not measured by the HSR1 and..." It seems like this light is measured by the HSR1, but it is just attributed to being direct irradiance?

Yes, this is correct. We have revised the text to clarify this point (P. 18, L304-308): "The HSR1 $F_{diffuse}$ is smaller than those from both MFRSRs, which may be partially related to the instrument design in how the HSR1 measures the $F_{diffuse}$ as noted previously (Badosa et al., 2014). This includes the isotropic assumption and the HSR1 wider FOV than the other instruments. In reality, some of the forward-scattered circumsolar radiation is included in the HSR1 $F_{direct}$ which would be measured as $F_{diffuse}$ by instruments with a narrower FOV. This explains much of the underestimation of $F_{diffuse}$ observed in this comparison study. "

Line 228 – Throughout the paper comparisons are done between the various reference instruments. As currently written, this seems unnecessary as I do not see what value it adds to the analysis, and it distracts from the main topic which is the evaluation of the HRS1.

The comparison results between the other instruments provide a reference for the HSR1 comparison. We have revised the text to highlight this point for why the other instruments are compared. For the irradiance, the text is revised to (P. 15, L261-263): "The MFRSR C1 and MFRSR E13 spectral irradiances are also compared to each other in Figs. 6-8 to provide context to the HSR1 comparison by showing the level of agreement between two instruments of the same model at the same location." For the AOD comparison, the text is revised to (P. 24, L429-430): "This comparison provides context to the HSR1 AOD comparison by quantifying the level of agreement between established instruments and AOD retrievals." We have also removed the tables and reduced the text to clarify the story.

Line 251 – In Figure 4 why not also include the direct irradiance?

We decided to focus on the total and diffuse since this is what is measured by the HSR1. The direct can be inferred by what is included in new Figure 8 (i.e., direct = total-diffuse). We have also plotted the direct irradiance comparison (blue) for your reference.

[Figure]

Line 255 – This table is hard to read and interpret and does not hold a lot of utility to the reader. Much of this information is already stated in the text, so I'd either omit the table or present the data in a graphical format.

Thank you for the suggestion. We have removed the tables. For the irradiance comparison, we have presented the correlation coefficients in the text as the values are nearly the same. For the AOD comparison, we have instead presented the correlation coefficients as the color shading in Figure 11a-h (see below).

[Figure]

Line 271-273 – "Similar to the total spectral irradiance, the MFRSR C1 and MFRSR E13 diffuse spectral irradiance comparison at 940 nm is the largest relative difference, which is nearly an order of magnitude larger than all other wavelengths (0.9-1.9%). This further highlights the challenges in measuring the spectral irradiance at 940 nm." I found this wording confusing and suggest it be revised.

We have revised the text for clarity (P. 20, L350-357): "Interestingly, the mean $F_{diffuse}$ for the HSR1 compared to those from the MFRSR C1 at 940 nm agree better than the MFRSR C1 and MFRSR E13 $F_{diffuse}$ at 940 nm of 9.8%. However, the mean differences for the $F_{diffuse}$ at 940 nm are small in magnitude at only 0.001-0.003 W m$^{-2}$ nm$^{-1}$. Similar to the $F_{total}$ comparison, the MFRSR C1 and MFRSR E13 $F_{diffuse}$ relative difference is largest at 940 nm compared to the relative differences at other MFRSR wavelengths. For context, the relative difference at 940 nm is nearly an order of magnitude larger than all other wavelengths (0.9-1.9%). In addition, the HSR1 $F_{diffuse}$ at 940 nm was within the MFRSR uncertainty of the MFRSRs by the same amount or more so (5-12%) than the MFRSRs were with each other (5.4%). This further highlights the challenges in measuring the spectral irradiance at 940 nm as two of the same instruments in the same location differ the most at this channel."

Line 279 – the details about the MFRSR spectral channel widths seems better suited for Section 2.

Thank you for the suggestion. We have moved the MFRSR narrowband filter details to Section 2.

Line 287 – what is the motivation for comparing HSR1 to SASHe under clear sky-conditions. As is this manuscript is only presenting statistical quantities of HSR1 versus other instruments with little justification for doing so or interpretation of the results. For example, how might the shadowmask design of HSR1 influence this comparison with a shadowband type instrument?

The SASHe comparison is limited to clear-sky due to SASHe data issues during this time period as discussed in Section 2.2.3. We have revised the SASHe data section to clarify this point.

We have added in text on how the shadowmask influences comparison with the shadowband systems (P. 33-34, L615-626): "The other distinctive feature is the low $F_{diffuse}$ measurement of the HSR1 relative to all the reference instruments. This was also noted by Badosa et al. (2014), and is a feature of the shading mask design. This low bias in $F_{diffuse}$ has several possible causes:

1. The wide FOV of the HSR1 optics compared to the narrower FOV of the MFRSR, which means that forward-scattered circumsolar radiation is excluded from the HSR1 $F_{diffuse}$ measurement, but included in the MFRSR measurement, which is able to measure the circumsolar component directly. Interestingly, the SASHe appears to show some similarities to the HSR1 in this regard. The circumsolar fraction increases with increasing AOD and cloud optical depth (COD), and hence, $F_{diffuse}$. Both SASHe and HSR1 show a reducing diffuse ratio with increasing diffuse irradiance, implying more of the circumsolar irradiance is included in $F_{direct}$ compared to the other references.

2. Manufacturing tolerances within the HSR1 shading mask may deviate from the assumption that the open areas are exactly 50% of the full hemisphere."

Line 304-312 – again, what about the instruments or experimental setup is driving these differences.

We plan to add in text on how the instrumental designs are potentially driving the differences seen in the revised discussion section (Sect. 5).

Line 325 – this sentence should be reworded: AODs are not collected, they are calculated.

We believe that this is a typo as the original text says that the AODs are collocated and not collected (Section 4.2). Regardless, we have removed the word "collocated" here.

Line 340 – again, I do not see the value in the comparison of the AOD derived from the reference instruments.

The comparison results between the other instruments provide a reference for the HSR1 comparison. We have revised the text to highlight this point for why the other instruments are compared. For the AOD comparison, the text is revised to (P. 24, L429-430): "This comparison provides context to the HSR1 AOD comparison by quantifying the level of agreement between established instruments and AOD retrievals." We have also removed the tables and reduced the text to clarify the story.

Line 375 – it may be worthwhile to include a timeseries figure or two of irradiance and AOD that illustrates under what solar zenith angle and cloud conditions there is good and poor agreement between HSR1 and the reference instruments.

Thank you for the suggestion. We have added in a new Figures 4 and 5 that is a timeseries of the irradiance(500 nm) and AOD(500 nm) with a clear-sky marker.

Line 423 – "Noting the measurement uncertainty of ±3% in the diffuse flux (Michalsky and Long, 2016), only 16.5% (all times) and 18.3% (clear-sky times) of the diffuse flux errors due to considering the HSR1 diffuse ratio are within measurement uncertainty." I had a hard time understanding this sentence and would recommend rewording it.

We have revised this sentence for clarity (P. 31, L570-572): "The measurement uncertainty of the $F_{broadband,diffuse}$ is ±3% (Sect. 2.2.4). If the $F_{broadband,diffuse}$ is determined by the $DR_{HSR1}$, then the $F_{diffuse, error}$ considering the $DR_{HSR1}$ are within the $F_{broadband,diffuse}$ measurement uncertainty only 16.5% (all times) and 18.3% (clear-sky times) of the time."

Line 509 – This section is not a discussion section but it is a summary. Here is a good place to discuss how the design of the HSR1 impacts its ability to measure irradiance relative to the reference instruments. Under what conditions does is perform well (e.g., clear-sky, cloudy-sky)? And when there are biases in the data HSR1 produces, why? For example, what impact does the wide field-of-view, the cosine response of the sensor, the assumption that the diffuse light is isotropic, etc., have on the measurements.

Thank you for the suggestions. We have expanded the discussion section to include further discussion on how the HSR1 performance and evaluation relates to instrument design in a revised version.

---

## Author Response (AR2)

Dear Dr. Schmidt,

Thank you for guiding the review of our manuscript entitled " Evaluation of the hyperspectral radiometer (HSR1) at the ARM SGP site" (doi: 10.5194/amt-2023-115).

We are now submitting a further revised manuscript in which we have addressed the second round of reviewer comments and suggestions, which helped to further improve the manuscript. The point-by-point responses are included below with the reviewer's comments in black and our replies in blue. The page and line numbers correspond to the change accepted version of the manuscript (i.e., "clean").

Thank you for your consideration of this manuscript.

Sincerely,
Dr. Kelly Balmes
* * *
Reviewer #1

The authors have done a very good job of responding to the comments from the previous review. However, I still have some minor comments, which are listed below.

Thanks for the positive review. We appreciate the reviewer's efforts to review this paper. See our replies below.

1. P3L79: "This is in contrast to rotating shadowband systems which must make the Ftotal and Fdiffuse measurements separately and, therefore, at different times." – Using two instruments, the two quantities are measured simultaneously.

We have revised the text to include the situation of two instruments measuring simultaneously as well (P. 3, L68-72): "Due to the nature of the measurements, the $F_{total}$ and $F_{diffuse}$ are measured simultaneously, and can be measured at a frequency up to 1 Hz. This is in contrast to rotating shadowband systems which must make the $F_{total}$ and $F_{diffuse}$ measurements at different positions of the shadowband rotation, and, therefore, at different times in the operating cycle. The simultaneously measured HSR1 $F_{total}$ and $F_{diffuse}$ is similar to two nearby instruments measuring $F_{total}$ and $F_{diffuse}$ separately but simultaneously."

2. P3L84: "spectrally disperses the light from the input fibres onto a 2D image sensor" – omit the word "light" which refers to a limited wavelength range. Better use "solar radiation" instead.

Thank you for the suggestion. We have revised the text by changing "light" to "solar radiation" (P. 3, L76).

3. Fig. 2: Use a legend for "Collocated Ftotal (gray) and Fdiffuse (pink) from the MFRSR C1 (square), MFRSR E13 (x-mark), and SASHe (circle)"

Thank you for the suggestion. We have revised Figure 2 by adding in a legend that includes MFRSRs and SASHe information. See revised Figure 2 below.

[Figure]

4. P7L77: "As with all spectrometers, measurements at the two extremes of the spectrum have low sensitivity and, therefore, additional noise is apparent." – The two extremes are referred to the edges of the spectrum, I guess. Please rephrase for clarification.

We have rephrased the text for clarity by changing "extremes" to "edges" (P. 7, L122).

5. P7L186: "The dome lensing effect corrected Ftotal and Fdiffuse are discussed further in Sect. 5." Please rephrase.

We have rephrased the text to (P. 7, L131-132): "The $F_{total}$ and $F_{diffuse}$ corrected for the dome lensing effect are discussed further in Sect. 5. These corrections will also be the subject of a future study as noted in Sect. 5."

6. P8L235: "A reference HSR1 is calibrated by removing the shading mask, and exposing the sensors to a 1000 W 'FEL' lamp" Does it mean, that the shading mask itself has no effect on the sensitivity?

The shading mask has no effect on the sensitivity to direct radiation, for those sensors which are not shaded by the mask. The mask geometry does however affect the sensitivity to diffuse radiation, as corrected for by the factor 2 in Eq. 1. This has been clarified by the added sentence (P. 8, L141-142): "This enables identical calibration of the seven sensors to direct beam light. The same sensitivity applies to diffuse light, though modified by the geometry of the shadowmask (Eq. 1)."

7. P8L243: "This means that the HSR1 Fdiffuse measurement will typically be lower than the corresponding measurements from a sun photometer or broadband tracker system." Please quantify.

We have added in a reference to the analysis by Norgren et al. (2022), which includes an analysis on the impact of the HSR1's field of view (P. 8-9, L150-153): "An analysis by Norgren et al. (2022) (see their Appendix A) quantified this for the case of thin clouds, estimating a circumsolar irradiance varying between negligible and ~10% of the direct beam, depending on solar zenith angle and cloud thickness. Implementing a correction for this will be a topic for further study."

8. P9L250: "The CSPHOT observations considered 250 include the AODs at 440, 500, 675, and 870 nm." Maybe add "wavelength" at the end.

Thank you for the suggestion. We have added in "wavelengths" to the clarify the text for the data descriptions of the CSPHOT (P. 9, L160), MFRSR (P. 9, L164 and L173), and SASHe (P. 9, L180).

9.  P9L274: The "Langley calibration" is mentioned here. Please give a reference.

The Langley calibration applied to SASHe is described in Appendix A and Flynn et al. (2016). We have revised the text to add in an additional reference to Appendix A (P. 10, L184).

10. Figure 5: Is not discussed in the text. Leave it out.

Thank you for the suggestion. We have removed Figure 5.

11. Chapter 4: In the fourth chapter, it was difficult to maintain attention. The reader could get tired of all the numbers. Is there a way to shorten it?

Thank you for the suggestion. We have revised Sect. 4 by shortening the text. We have shortened it by combining MFRSR C1 and E13 comparison results when similar. In addition, we have reduced numbers corresponding to correlation coefficient and regression slope values.

12. Appendix A: Is this needed? Giving a reference would be sufficient.

We have included Appendix A since there is currently no reference on the SASHe except for a technical report (Flynn, 2016). This appendix helps provides more instrument details on the SASHe if the reader is interested.

13. Appendix B: As written in my previous review, an extension to PAR is not necessary to emphasize the instrumental capabilities.

We agree that the PAR comparison is not necessary to emphasize the instrument's capabilities. However, the PAR comparison shows an application of the HSR1 available with hyperspectral radiometers. Therefore, we have placed the PAR comparison in the Appendix instead of the main text.

---

## Author Response (AR3)

Dear Dr. Schmidt,

Thank you for guiding the review of our manuscript entitled " Evaluation of the hyperspectral radiometer (HSR1) at the ARM SGP site" (doi: 10.5194/amt-2023-115).

We are now submitting a further revised manuscript in which we have addressed your comments and suggestions, which helped to further improve the manuscript. The point-by-point responses are included below with your comments in black and our replies in blue. The page and line numbers correspond to the change accepted version of the manuscript (i.e., "clean").

Thank you for your consideration of this manuscript.

Sincerely,
Dr. Kelly Balmes
* * *
Dear authors,
thank you very much for submitting this revised version. I only have some technical changes at this point.

Thanks for the positive review. We appreciate your efforts to review and facilitate the review of this paper. See our replies below.

* Figure 3 (caption): "Heavy blue line" --> "The heavy blue line"

Thank you for catching this. We have revised the caption of Figure 3 (P. 8, L137) from "Heavy blue line" to "The heavy blue line."

* Line 555 in track-change manuscript: "The exception is the MFRSR AODs compared to the SASHe with larger regression slopes and positive bias slopes." This is a bit unclear. I think somewhere a "that" or "where" is missing. Also, please make sure that it is clear what the exception is from (I believe it relates to the previous sentence).

We have revised the text to (P. 22, L407-409): "The exception is the MFRSR AODs compared to the SASHe where the regression slopes are larger than 1 instead of near 1 and the bias regression slopes are positive instead of negative."

* Line 816: The HSR1 and SASHe derived AODs vary slightly more compared to each other --> do you mean differ instead of vary, or really vary? Vary from each other does not seem to make as much sense as differ from each other.

Thank you for the suggestion. We have revised the text (P. 32, L599) by changing "vary" to "differ."

* Line 880: in diffuse irradiance, that is wavelength dependent --> either remove comma (recommended), or replace 'that' with 'which'

Thank you for the suggestion. We have revised the text by removing the comma (P. 34, L656).

Regarding figures A1, 1: for the next revision, please check if your figures containing maps/aerial images and photos require a copyright statement/image credit and add it to the figures (or captions) (https://publications.copernicus.org/for_authors/manuscript_preparation.html#mapsaerials, https://publications.copernicus.org/for_authors/manuscript_preparation.html#figurestables -> Reproduction and reuse of figures and tables). If these figures were entirely created by the authors, there is no need to add a copyright statement or credit. In that case it is important that you confirm this explicitly by email.

We have added in a reference to Google Earth for the map image in the caption for Figure 1. In addition, Figure A1 is made by an author (Connor Flynn). We will confirm this by email explicitly.